# Fractal cycles of sleep, a new aperiodic activity-based definition of sleep cycles

Yevgenia Rosenblum[1]*, Mahdad Jafarzadeh Esfahani[1], Nico Adelhöfer[1], Paul Zerr[1], Melanie Furrer[2], Reto Huber[2,3], Famke F Roest[1], Axel Steiger[4], Marcel Zeising[5], Csenge G Horváth[6], Bence Schneider[6], Róbert Bódizs[6], Martin Dresler[1]

[1]Radboud University Medical Centre, Donders Institute for Brain, Cognition and Behavior, Nijmegen, Netherlands; [2]Child Development Center and Children's Research Center, University Children's Hospital Zürich, University of Zürich, Zürich, Switzerland; [3]Department of Child and Adolescent Psychiatry and Psychotherapy, Psychiatric University Hospital Zurich, Zurich, Switzerland; [4]Max Planck Institute of Psychiatry, Munich, Germany; [5]Klinikum Ingolstadt, Centre of Mental Health, Ingolstadt, Germany; [6]Semmelweis University, Institute of Behavioural Sciences, Budapest, Hungary

*For correspondence:
yevgeniabio@yahoo.com

Competing interest: The authors declare that no competing interests exist.

## eLife Assessment

This **valuable** study provides a novel method to detect sleep cycles based on variations in the slope of the power spectrum from electroencephalography signals. The method, dispensing with time-consuming and potentially subjective manual identification of sleep cycles, is supported by **solid** evidence and analyses. This study will be of interest to researchers and clinicians working on sleep and brain dynamics.

**Abstract** Sleep cycles are defined as episodes of non-rapid eye movement (non-REM) sleep followed by an episode of REM sleep. Fractal or aperiodic neural activity is a well-established marker of arousal and sleep stages measured using electroencephalography. We introduce a new concept of 'fractal cycles' of sleep, defined as a time interval during which time series of fractal activity descend to their local minimum and ascend to the next local maximum. We assess correlations between fractal and classical (i.e. non-REM – REM) sleep cycle durations and study cycles with skipped REM sleep. The sample comprised 205 healthy adults, 21 children and adolescents and 111 patients with depression. We found that fractal and classical cycle durations (89±34 vs 90±25 min) correlated positively (r=0.5, p<0.001). Children and adolescents had shorter fractal cycles than young adults (76±34 vs 94±32 min). The fractal cycle algorithm detected cycles with skipped REM sleep in 91–98% of cases. Medicated patients with depression showed longer fractal cycles compared to their unmedicated state (107±51 vs 92±38 min) and age-matched controls (104±49 vs 88±31 min). In conclusion, fractal cycles are an objective, quantifiable, continuous and biologically plausible way to display sleep neural activity and its cycles.

## Introduction

The cyclic nature of sleep has long been established with a classical sleep cycle defined as a time interval that consists of an episode of non-rapid eye movement (non-REM) sleep followed by an episode of REM sleep (*Feinberg and Floyd, 1979*; *Le Bon, 2020*). Typically, nocturnal sleep consists

of 4–6 such cycles, which last for about 90 min each. Every cycle is seen as a fundamental physiological unit of sleep central to its function (*Feinberg, 1974*) or a miniature representation of the sleep process (*Le Bon et al., 2002*).

Basic structural organization of normal sleep is rather conservative with some exceptions. Thus, occasionally, at the beginning of the night in healthy adolescents and young adults, there could occur cycles with skipped REM sleep, which are also called 'skipped' cycles. In skipped cycles, a REM sleep episode is expected to appear except that it does not and only a 'lightening' of sleep is observed presumably due to too high non-REM pressure (*Le Bon, 2020*). Likewise, some alterations of the sleep structure can be observed in sleep disorders, for example, narcolepsy and insomnia (*Scammell, 2015*), and healthy aging (*Carrier et al., 2011*; *Conte et al., 2014*). In some neurological and psychiatric conditions, such as major depressive disorder (MDD), Parkinson's and Alzheimer's diseases, sleep architecture disturbances are further linked to the disease neuropathology (*Courtet and Olié, 2012*; *Palagini et al., 2013*; *Pillai and Leverenz, 2017*).

While the importance of sleep cycles is indisputable, their function as a unit is poorly understood and surprisingly under-explored, especially when compared to the extensive research on sleep stages (either non-REM or REM) or sleep microstructure (e.g. sleep spindles, slow waves, microarousals). One of the reasons for this striking absence of research progress might be the lack of a proper quantifiable and reliable objective measure from which sleep cycles could be derived directly (*Schneider et al., 2022*).

Currently, sleep cycles are defined via a visual inspection of the hypnogram, the graph in which categorically separated sleep stages are plotted over time. Yet assigning a discrete category to each sleep stage is rather arbitrary as sleep stages are presumably continuous and thus do not occur as steep lines of a hypnogram. In addition, visual sleep stage scoring is very time-consuming, subjective and error-prone with a relatively low (~80%) inter-rater agreement. This results in a low accuracy regarding the sleep cycle definition.

We suggest that a data-driven approach based on a real-valued neurophysiological metric (as opposed to the categorical one) with a finer quantized scale could forward the understanding of sleep cycles considerably. Specifically, we propose that research on sleep cycles would benefit from recent advances in the field of fractal neural activity. In literature, fractal activity is also called aperiodic, non-oscillatory, 1 /f or scale-free activity, being named after the self-similarity exhibited by patterns of sensor signals across various time scales. Fractal activity is a distinct type of brain dynamics, which is sometimes seen as a 'background' state of the brain, from which linear, rhythmic (i.e. periodic, oscillatory) dynamics emerge to support active processing (*Buzsaki, 2006*; *Freeman, 2006*). Growing evidence confirms that fractal activity has a rich information content, which opens a window into diverse neural processes associated with sleep, cognitive tasks, age, and disease (*Voytek and Knight, 2015*; *Bódizs et al., 2024*; *Höhn et al., 2024*).

Fractal dynamics follow a power-law 1 /f function, where power decreases with increasing frequency (*He, 2014*). The steepness of this decay is approximated by the spectral exponent, which is equivalent to the slope of the spectrum when plotted in the log-log space (*He, 2014*; *Gerster et al., 2022*). The fractal signal is not dominated by any specific frequency, rather it reflects the overall frequency composition within the time series (*Horváth et al., 2022*) such that steeper (more negative) slopes indicate that the spectral power is relatively stronger in slow frequencies and relatively weaker in faster ones (*He, 2014*).

In terms of mechanisms, it has been suggested that flatter high-band (30–50 Hz) fractal slopes reflect a shift in the balance between excitatory and inhibitory neural currents in favour of excitation while steeper slopes reflect a shift towards inhibition (*Gao et al., 2017*). Given that the specific balance between excitation and inhibition defines a specific arousal state and the conscious experience of an organism (*Nir and Tononi, 2010*), the introduction of Gao's model led to an increased interest in fractal activity. For example, it has been shown that high-band fractal slopes discriminate between wakefulness, non-REM and REM sleep stages as well as general anesthesia or unconsciousness (*Gao et al., 2017*; *Colombo et al., 2019*; *Lendner et al., 2020*; *Höhn et al., 2024*).

Of note, Gao's model does not account for the lower part of the spectrum, which is also scale-free. An alternative model suggests that the broadband 1 /f² activity reflects the tendency of the central nervous system to alternate between UP- (very rapid spiking) and DOWN- (disfacilitation, no activity) states (*Milstein et al., 2009*; *Baranauskas et al., 2012*). Empirical studies further showed that the

broadband (2–48 Hz) slope is an especially strong indicator of sleep stages and sleep intensity with low inter-subject variability and sensitivity to age-related differences (*Miskovic et al., 2019*; *Schneider et al., 2022*; *Horváth et al., 2022*). Taken together, this literature suggests that fractal slopes can serve as a marker of arousal, sleep stages and sleep intensity (*Lendner et al., 2020*; *Schneider et al., 2022*; *Horváth et al., 2022*). We expect that this line of inquiry could be extended to sleep cycles.

On a related note, the reciprocal interaction model of sleep cycles assumes that each sleep stage involves distinct activation patterns of inhibitory and excitatory neural networks (*Pace-Schott and Hobson, 2002*). This model explains alternations between non-REM and REM sleep stages by the interaction between aminergic and cholinergic neurons of the mesopontine junction (*Pace-Schott and Hobson, 2002*). Notably, during REM sleep, acetylcholine plays a major role in maintaining brain activation, which is expressed as EEG desynchronization, one of the main features of REM sleep (*Nir and Tononi, 2010*). This is of special importance in affective disorders since according to one of the pathophysiological explanations of depression, for example, the cholinergic-adrenergic hypothesis, central cholinergic factors play a crucial role in the aetiology of affective disorders, with depression being a disease of cholinergic dominance (*Janowsky et al., 1972*). Many antidepressants (e.g. serotonin-norepinephrine reuptake inhibitors, selective serotonin reuptake inhibitors) suppress REM sleep and thus cause essential alterations in sleep architecture. Intriguingly, REM sleep suppression is related to the improvement of depression during pharmacological treatment with antidepressants enhancing monoaminergic neurotransmission (*Vogel et al., 1990*; *Wichniak et al., 2013*).

Based on this background, we propose that a fractal neural activity-based definition of sleep cycles has the potential to considerably advance our understanding of the cyclic nature of sleep, for example, by introducing graduality to the categorical concept of sleep stages. The current study analyzes the dynamics of nocturnal fluctuations in fractal activity using five independently collected polysomnographic datasets overall comprising 205 recordings from healthy adults. Based on the inspection of fractal activity across a night, we introduce a new concept of fractal activity-based cycles of sleep or 'fractal cycles' for short. We describe differences and similarities between fractal cycles defined by our algorithm and classical (non-REM – REM) cycles defined by the hypnogram. We hypothesize that the timing and durations of the fractal cycles would closely correspond to those of classical cycles. We had no prior hypothesis regarding correspondence between the fractal cycles and classical cycles with skipped REM sleep, that is this analysis was exploratory.

Given the above-mentioned age-related changes in fractal activity (flatter slopes) and sleep structure (fewer and shorter classical cycles), we also study whether fractal cycle characteristics change with age. To this end, we use 5 healthy adult datasets with the age range of 18–75 years (n=205). Moreover, we add to our study a pediatric polysomnographic dataset (age range: 8–17 years, n=21) to explore fractal cycles in childhood and adolescence, a life period accompanied by deepest sleep and massive brain reorganization (*Kurth et al., 2012*) as well as a higher frequency of cycles with skipped REM sleep (*Jenni and Carskadon, 2004*).

Finally, we test the clinical value of the fractal cycles by analyzing polysomnographic data in 111 patients with MDD, a condition characterized by disturbed sleep structure (besides its clinical symptoms, such as abnormalities of mood and affect). Specifically, we compare fractal cycles of sleep between medicated MDD patients (three MDD datasets, n=111) and healthy age-matched controls (n=111) as well as in the unmedicated and medicated states within the same MDD patients (one of the three MDD datasets, n=38). We hypothesize that the fractal cycle approach would be more sensitive in detecting differences between typical and atypical sleep architecture compared to the conventional classical cycles.

## Results
### Fractal cycles in healthy adults

*Figure 1A* displays smoothed fractal slope time series and hypnogram for an example subject. Four additional examples are presented in *Figure 1—figure supplement 1*. Fractal slope time series and hypnograms for all healthy adult participants are shown in Supplementary PowerPoint File shared on https://osf.io/gxzyd.

We observed that the slopes of the fractal (aperiodic) power component fluctuate across a night such that the peaks of the time series largely coincide with REM sleep episodes while the troughs of

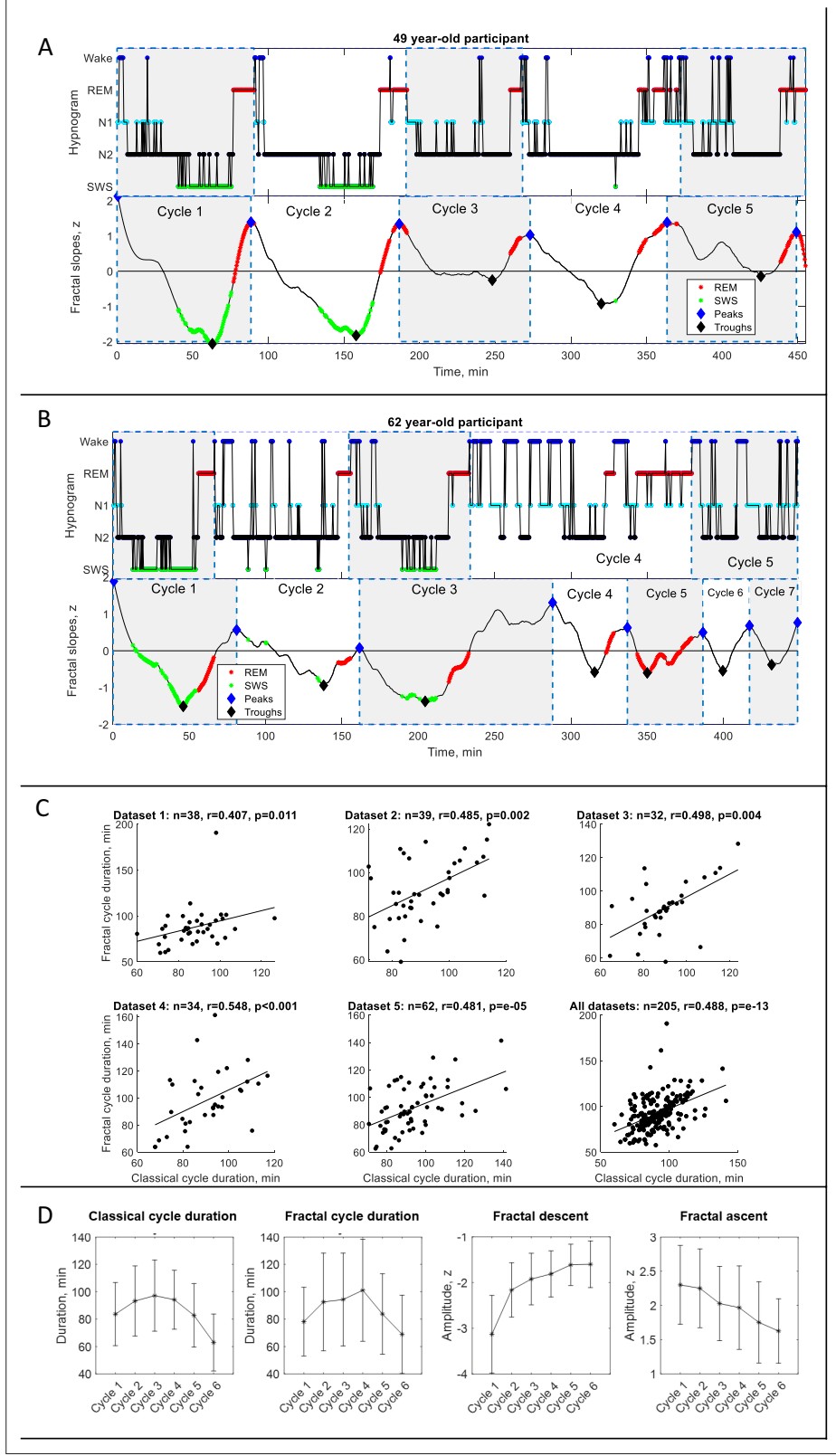

**Figure 1.** Fractal cycles in healthy adults. (**A – B**) Individual fractal and classical sleep cycles. Time series of smoothed z-normalized fractal slopes (bottom) and corresponding hypnograms (top) observed in two different participants. The duration of the fractal cycle is a time interval between two successive peaks (blue diamonds). (**A**) S15 from Dataset 3 shows a one-to-one match between fractal cycles defined by the algorithm and classical

*Figure 1 continued*

(non-REM – REM) cycles defined by the hypnogram. (**B**) In S22 from dataset 5, the second part of night has many wake epochs, some of them are identified by the algorithm as local peaks. This results in a higher number of fractal cycles as compared to the classical ones and a poor match between the fractal cycles No. 3–7 and classical cycles No. 2–5. The algorithm does not distinguish between the wake and REM-related fractal slopes and can define both as local peaks. Since the duration of the fractal cycles is defined as an interval of time between two adjacent peaks, more awakenings/arousals during sleep (usually associated with aging) are expected to result in more peaks and, consequently, more fractal cycles, that is a shorter cycle duration. This is one of the possible explanations for the correlation between the fractal cycle duration and age (shown in *Figure 1—figure supplement 4A*). Time series of the fractal slopes and corresponding hypnograms for all participants are reported in Supplementary PowerPoint File shared on https://osf.io/gxzyd. SWS – slow-wave sleep, REM – rapid eye movement. (**C**) Scatterplots: each dot represents the duration of the cycles averaged over one participant. The durations of the fractal and classical sleep cycles averaged over each participant correlate in all analyzed datasets, raw (non-ranked) values are shown, r – Spearman's correlation coefficient. (**D**) Cycle-to-cycle overnight dynamics show an inverted U shape of the duration of both fractal and classical cycles across a night and a gradual decrease in absolute amplitudes of the fractal descents and ascents from early to late cycles.

The online version of this article includes the following figure supplement(s) for figure 1:

**Figure supplement 1.** Individual fractal and classical sleep cycles in healthy adults.

**Figure supplement 2.** Fractal and classical cycles: distributions, means, correlations and an individual example.

**Figure supplement 3.** Individual cycles with skipped REM sleep.

**Figure supplement 4.** Correlations.

---

the time series for the most part coincide with non-REM sleep episodes. Based on this observation we propose the following definition:

Definition: The fractal activity-based cycles of sleep or 'fractal cycles' for short is a time interval during which the time series of the fractal slopes descend from the local maximum to the local minimum with the amplitudes higher than $|0.9|$ z, and then lead back from that local minimum to the next local maximum.

Based on this definition, we created an algorithm, which automatically defined the onset and offset of the fractal cycles (the adjacent peaks of the time series of the fractal slopes; available on https://osf.io/gxzyd/). An additional visual inspection showed that the automatic definition of fractal cycles (*Figure 1A*, blue diamonds) was identical to that provided by a human scorer.

Overall, fractal slopes cyclically descend and ascend 4–6 times per night and the average duration of such a descent-ascent cycle is close to 90 min. *Figure 1—figure supplement 2A* shows the frequency distribution of the fractal cycle durations for each dataset separately as well as for the pooled dataset.

This observation strikingly resembles what we know about classical sleep cycles: 'night sleep consists of 4–6 sleep cycles, which last for about 90 min each' (*Feinberg and Floyd, 1979*; *Le Bon, 2020*; *Figure 1—figure supplement 2A*, bottom panel). Further calculations showed that the mean duration of the fractal cycles averaged over all cycles from all datasets (n=940) is 89±34 min while the mean duration of the classical sleep cycles is 90±25 min (*Figure 1—figure supplement 2B*). The mean durations of the fractal and classical sleep cycles averaged over each participant correlated in all analyzed datasets (r=0.4–0.5, *Table 1*, *Figure 1C*).

*Figure 1—figure supplement 1D* shows fractal activity across 13 hr, including 3 hr before the sleep onset and 2 hr after awakening (using data from *Rosenblum et al., 2024b*). The pattern of fractal fluctuations suggests that fractal cycles are specific to sleep and are not observed during wake.

Cycle-to-cycle overnight dynamics showed an inverted U-shape of the fractal cycle durations and a gradual decrease in absolute amplitudes of the fractal descents and ascents from early to late cycles. This pattern resembled an inverted U-shape of the classical cycle durations (*Figure 1D*).

## Correspondence between fractal and classical cycles

Analysis at the individual cycle level revealed that 81% (763/940) of all fractal cycles (77–88% in different datasets) could be matched to a specific classical cycle defined by hypnogram, that is, the timings of fractal and classical cycles approximately coincide. Bayesian prevalence analysis further revealed that the Bayesian highest posterior density interval with 96% probability level lies within the

**Table 1.** Demographic, sleep and fractal characteristics of healthy adults.

| Characteristic | Dataset 1 | Dataset 2 | Dataset 3 | Dataset 4 | Dataset 5 | Pooled dataset |
|---|---|---|---|---|---|---|
| No. participants analyzed | 38 | 39 | 32 | 34 | 62 | 205 |
| Age, years | 46.8±10.7 | 31.0±9.9 | 45.3±15.9 | 21.5±3.8 | 37.4±15.3 | 36.7±15.0 |
| Age range, years | 29–65 | 19–54 | 22–75 | 18–35 | 20–66 | 18–75 |
| Sex, female, % | 53 | 54 | 61 | 68 | 55 | 58 |
| Wake, % | 6.0 | 4.9 | 7.5 | 7.1 | 9.1 | 7.0 |
| Non-REM stage 1, % | 7.7 | 11.9 | 9.0 | 3.6 | 7.5 | 7.9 |
| Non-REM stage 2, % | 48.1 | 45.9 | 49.3 | 34.7 | 46.1 | 45.1 |
| Slow-wave sleep, % | 19.2 | 20.3 | 16.2 | 34.2 | 17.2 | 20.9 |
| REM sleep, % | 19.0 | 16.9 | 17.9 | 19.3 | 19.3 | 18.6 |
| Total sleep time, min | 394±55 | 430±26 | 434±37 | 445±62 | 467±38 | 438±51 |
| Classical sleep cycle duration, min | 86.2±23.3 | 90.0±21.3 | 89.0±22.7 | 92.2±23.7 | 91.9±29.0 | 90.1±24.9 |
| Fractal sleep cycle duration, min | 86.4±35.2 | 90.0±25.5 | 86.4±31.2 | 94.7±37.1 | 89.9±37.1 | 89.1±34.0 |
| Classical-fractal cycles duration correlation, r | 0.407 | 0.485 | 0.498 | 0.548 | 0.481 | 0.488 |
| Classical-fractal cycles duration correlation, p | 0.011 | 0.002 | 0.004 | 0.001 | $10^{-5}$ | $10^{-13}$ |
| One-to-one match between classical and fractal cycles timing and duration, % cycles | 78 | 88 | 82 | 87 | 77 | 81 |
| Participants having all fractal and classical cycles in a one-to-one match, % participants | 53 | 62 | 66 | 53 | 45 | 54 |
| Descent amplitude, z | –2.2±0.9 | –2.3±0.9 | –2.2±0.8 | –2.2±0.8 | –2.1±0.8 | –2.2±0.8 |
| Ascent amplitude, z | 2.1±0.6 | 2.2±0.6 | 2.1±0.6 | 2.1±0.6 | 2.0±0.6 | 2.2±0.6 |
| No. fractal cycles | 167 | 171 | 152 | 152 | 298 | 940 |
| No. classical cycles | 171 | 180 | 146 | 161 | 303 | 961 |
| No. 'skipped' first cycles (%) | 5 (13%) | 7 (18%) | 1 (3%) | 19 (56%) | 15 (24%) | 47 (23%) |

‡shows mean and SD, r – Spearman's correlation coefficient, 'skipped' cycle – a cycle where REM sleep does not appear, REM – rapid eye movement.

0.77–0.83 range (the range within which the true population value lies) and the maximum a posteriori point estimate prevalence is equal to 0.8, reflecting the most likely values for the population parameter. This analysis reflects the within-participant replication probability: the probability of obtaining a significant experimental result if the same experiment was applied to a new participant randomly selected from the population (*Ince et al., 2022*).

**Table 2.** Sources of fractal and classical cycle mismatches.

| Characteristic | Dataset 1 (A) | Dataset 2 (B) | Dataset 3 (C) | Dataset 4 | Dataset 5 | Pooled dataset |
|---|---|---|---|---|---|---|
| Classical – fractal cycle duration difference, min | 13.2±15.9 | 9.6±9.1 | 8.0±11.3 | 13.0±17.0 | 11.9±10.2 | 11.3±12.7 |
| WASO, % | 6.0±5.6 | 4.9±3.6 | 7.5±5.0 | 7.1±4.2 | 9.1±5.7 | 7.0±5.2 |
| WASO %, r | –0.011 | 0.488 | 0.377 | 0.141 | 0.361 | 0.226 |
| WASO %, p | 0.950 | 0.002 | 0.034 | 0.425 | 0.004 | 0.001 |
| Descent amplitude, z | –2.3±0.9 | –2.5±0.9 | –2.3±0.8 | –2.0±0.7 | –2.1±0.8 | –2.2±0.8 |
| Fractal descent, r | 0.189 | 0.327 | 0.143 | 0.144 | 0.149 | 0.152 |
| Fractal descent, p | 0.171 | 0.002 | 0.182 | 0.271 | 0.135 | 0.002 |
| Ascent amplitude, z | 2.3±0.6 | 2.1±0.5 | 2.2±0.6 | 2.0±0.6 | 2.0±0.6 | 2.1±0.6 |
| Fractal ascent, r | 0.109 | –0.105 | –0.103 | 0.028 | –0.010 | –0.062 |
| Fractal ascent, p | 0.432 | 0.318 | 0.339 | 0.835 | 0.918 | 0.217 |
| Skipped cycle lengths/TST, proportion | 0.144 | 0.223 | 0.139 | 0.249 | 0.201 | 0.206 |
| Skipped cycle lengths/TST, r | 0.098 | –0.363 | 0.384 | –0.216 | 0.374 | –0.019 |
| Skipped cycles length/TST, p | 0.788 | 0.303 | 0.523 | 0.334 | 0.066 | 0.873 |
| REM episode length, min | 23.5±15.2 (72 cycles) | 22.8±13.2 (93 cycles) | 21.8±11.6 (90 cycles) | 26.0±13.9 (60 cycles) | 24.3±15.0 (102 cycles) | 0.251±0.08 (417 cycles) |
| REM episode length, r | 0.222 | 0.411 | 0.400 | 0.231 | 0.394 | 0.358 |
| REM episode length, p | 0.061 | <0.001 | 0.001 | 0.076 | <0.001 | <0.001 |

All parameters listed in the first column were correlated with the absolute value of the difference in classical vs fractal sleep cycle durations. For WASO and skipped cycles, all cycles of a given participant were averaged and the correlations were performed at the subject level. For the rest of the parameters, fractal and classical cycles were matched one-to-one when possible (~50% of all participants) and correlations were performed at the cycle level, r's higher than 0.7 are considered as strong correlation scores, values lower than 0.3 are considered as weak, r's values in the range of 0.3–0.7 are considered as moderate scores.

REM – rapid eye movement sleep, WASO – wake after sleep onset, TST – total sleep time, r – Spearman correlation coefficients.

In 54% (111/205) of the participants (45–66% in different datasets), all fractal cycles approximately coincided with classical cycles (r=0.5–0.8, p<0.001, *Table 1* and *Figure 1—figure supplement 2C*). Bayesian prevalence analysis revealed that the maximum a posteriori point estimate prevalence is equal to 0.52 and the Bayesian highest posterior density interval (the true population level) with 96% probability level lies within the 0.45–0.60 range.

In the remaining 46% of the participants, the difference between the fractal and classical cycle numbers ranged from –2 to 2 with the average of –0.23±1.23 cycle. This subgroup had 4.6±1.2 fractal cycles per participant, while the number of classical cycles was 4.9±0.7 cycles per participant. The correlation coefficient between the fractal and classical cycle numbers was 0.280 (p=0.006) and between the cycle durations – 0.278 (p=0.006). Still, in these participants, many – even though not all – fractal cycles could be matched to a specific classical cycle. *Figure 1B* displays such an example in one participant. More examples can be found in *Figure 1—figure supplement 1C, D* and Supplementary PowerPoint File shared on https://osf.io/gxzyd.

## Sources of fractal and classical cycle mismatches

The timings and correlations between the fractal and classical cycles were not one-to-one (r=0.6–0.8, p<0.001). We identified two possible sources of a mismatch (*Table 2*; see also Table 5).

### REM episode duration

While the fractal cycle end is defined as the local maximum of time series of fractal slopes, the classical cycle ends with the end of a REM episode. As a consequence, in some cases, especially for morning cycles that have rather long REM periods (>20 min), the match between fractal and classical cycles can

be rather coarse-grained (See, for example, cycle 3 in S16, *Figure 1—figure supplement 1A*). Yet, in other cases, the match between fractal and classical cycles might be almost perfect (See *Figure 1A*).

To test this visual observation, we correlated the absolute values of the difference in classical vs fractal sleep cycle durations with the REM episode length within a given cycle. We included in this analysis only the participants who had an equal number of fractal and classical cycles in order to match each fractal cycle to a classical cycle individually. We found that longer REM episodes were associated with a higher difference between classical vs fractal sleep cycle durations ($r$=0.36, p<0.001, n=417 cycles, *Table 2*). Interestingly, our recent study has shown that fractal activity within a REM sleep episode is not homogenous, with phasic states showing steeper fractal slopes than tonic ones (*Rosenblum et al., 2025*).

## Wake after sleep onset (WASO) duration

Visual inspection of the data suggested that participants with more WASO often had more fractal than classical cycles. This might stem from the fact that both REM- and wake-related smoothed fractal slopes could be defined as local peaks (*Figure 1A, B*, *Figure 4—figure supplement 4*). More fractal peaks imply more fractal cycles and thus, possibly, more mismatches between the number and duration of classical and fractal cycles. To test this hypothesis, we correlated the average difference between the durations of classical and fractal cycles for each participant with the WASO proportion. We found that a higher difference in cycle durations was associated with a higher WASO proportion in 3/5 datasets (r's=0.36–0.49, p<0.030) as well as in the merged dataset ($r$=0.23, p=0.001, n=205 participants, *Table 2*).

In addition, we correlated the difference in classical vs fractal cycle durations with the fractal descent or ascent amplitudes (as reflections of fractal cycle depth and possibly sleep quality). We found that a shallower fractal descent was associated with a higher mismatch between fractal and classical cycles in 1/5 datasets ($r$=0.33, p=0.02) as well as in the merged dataset ($r$=0.15, p=0.002, n=400 cycles, *Table 2*).

## Fractal cycles in children and adolescents

Next, we explored fractal cycles in children and adolescents (mean age: 12.4±3.1 years, n=21, *Appendix 1—table 3*) and compared them with those in young adults (mean age: 24.8±0.9 years, n=24). Two examples of smoothed fractal slope time series and hypnograms from the pediatric dataset are shown in *Figure 2A* – B. All examples are shown in Supplementary PowerPoint File shared on https://osf.io/gxzyd.

We found that children and adolescents had shorter fractal cycles compared to young adults with a medium effect size (76±34 vs 94±32 min, p<0.001, Cohen's d=–0.57, 112 vs 121 pooled cycles, 5.0 cycles/participant vs 4.4 cycles/participant, *Figure 2C, D*, *Appendix 1—table 3*). Similarly, children and adolescents showed shorter classical cycles than young adults with a medium effect size (80±23 vs 90±22 min, p<0.001, Cohen's d=–0.42, 112 vs 114 pooled cycles, *Figure 2C, D*).

To directly compare the fractal and classical approaches, we performed a Multivariate Analysis of Variance with fractal and classical cycle durations as dependent variables, the group as an independent variable and the age as a covariate. We found that fractal cycle durations showed higher F-values ($F_{(1, 43)}$=4.5 vs $F_{(1, 43)}$=3.1), adjusted R squared (0.138 vs 0.089) and effect sizes (partial eta squared 0.18 vs 0.13) than classical cycle durations.

Cycle-to-cycle overnight dynamics further revealed that the first and second fractal – but not classical – cycles were significantly shorter in the pediatric compared to the control group (*Figure 2E*) with medium effect sizes (d=–0.61–0.72). At the same time, the overnight classical – but not fractal – cycle analysis detected a between-group difference for the fourth classical cycle with a large effect size (d=–1.0, *Figure 2E*).

## Skipped cycles

We tested whether the fractal cycle algorithm can detect skipped cycles, that is the cycles where an anticipated REM episode is skipped possibly due to too high homeostatic non-REM pressure. We counted only the first classical cycles (i.e. the first cycle out of the 4–6 cycles that each participant had per night, *Figure 2A* – B) as these cycles coincide with the highest non-REM pressure. An additional reason to disregard skipped cycles observed later during the night was our aim to achieve higher

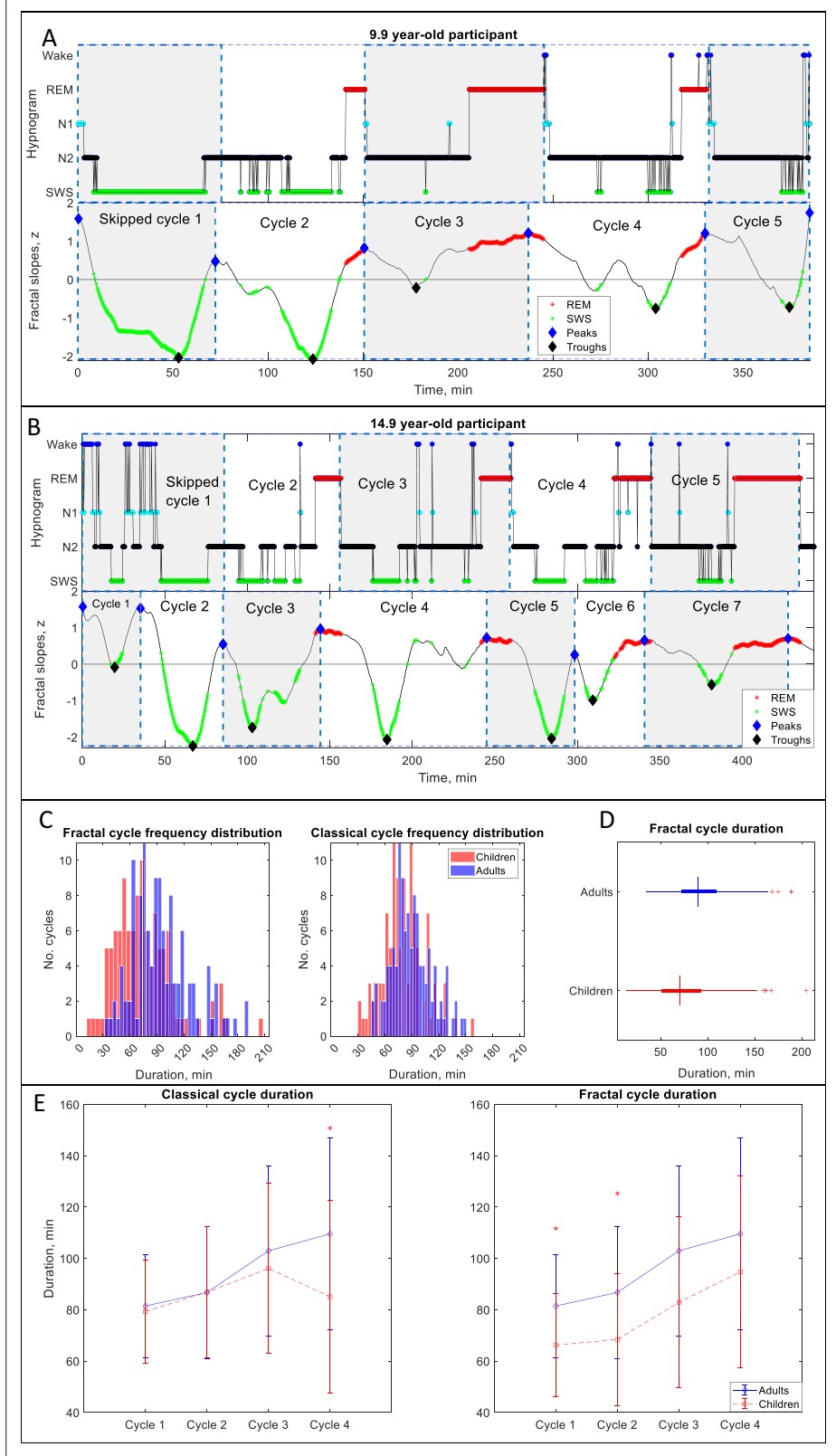

**Figure 2.** Fractal cycles in children and adolescents. (**A, B**) Individual cycles: time series of smoothed z-normalized fractal slopes (bottom) and corresponding hypnograms (top). The duration of the fractal cycle is a time interval between two successive peaks (blue diamonds) defined with the Matlab function *findpeaks* with a minimum peak distance of 20 min and minimum peak prominence of 0.9 z. SWS – slow-wave sleep, REM – rapid eye

*Figure 2 continued*

movement sleep. (**A**) In this 9.9-year-old participant (from Dataset 6), we split the first 150-min-long classical cycle into two cycles according to the definitions of a 'skipped' cycle presented in Materials and methods. The fractal cycle algorithm successfully detected this skipped cycle. (**B**) This 14.9-year-old participant has a 156-min-long first classical cycle. Visual inspection shows that it should be divided into 3 skipped cycles, however, our a priori definition of skipped cycles did not include an option to subdivide a long cycle into three short cycles; hence, we split it into two short cycles. The fractal cycle algorithm was sensitive to these sleep lightenings and detected all three short cycles. Classical cycle 4 looks like a skipped cycle as it has two clear episodes of slow-wave sleep separated by non-REM stage 2. However, the length of this cycle is shorter than 110 min (the threshold defined a priori), therefore, we did not split the classical cycle 4 into two cycles. The fractal cycle algorithm was sensitive to this lightening of sleep and defined two fractal cycles during this period. (**C**) Histograms: The frequency distribution of fractal (left) and classical (right) cycle durations in children and adolescents (mean age: 12.4±3.1 years) compared to young adults (mean age: 24.8±0.9 years). Kolmogorov-Smirnov's test rejected the assumption that cycle duration comes from a standard normal distribution. (**D**) Box plots: in each box, a vertical central line represents the median, the left and right edges of the box indicate the 25th and 75th percentiles, respectively, the whiskers extend to the most extreme data points not considered outliers, and a plus sign represents outliers. Children and adolescents show shorter fractal cycle duration compared to young adults. (**E**) Overnight dynamics: cycle-to-cycle dynamics show that the first and the second fractal cycles are shorter in the pediatric compared to control group, * marks a statistically significant difference between the groups.

between-subject consistency as second – sixth skipped cycles were observed in only a small number of participants and were not distributed equally across the datasets.

The average number of the first skipped cycles for Datasets 1–5 is reported in *Table 1*. *Appendix 1—table 9* further reports the average number of skipped cycles as assessed by two independent raters and the inter-rater agreement. Three specific examples of skipped cycles in young adults are presented in *Figure 1—figure supplement 3* and two examples in children are shown in *Figure 2A, B*. All cycles are marked in Supplementary PowerPoint File shared on https://osf.io/gxzyd.

Visual inspection of the hypnograms from Datasets 1–6 was performed by two independent researchers. Scorer 1 and Scorer 2 detected that out of 226 first sleep cycles 58 (26%) and 64 (28%), respectively, lacked REM episodes. The agreement on the presence of skipped cycles between two human raters equaled 91% (58 cycles detected by both raters out of 64 cycles detected by two scorers). The fractal cycle algorithm detected skipped cycles in 57 out of 58 (98%) cases detected by Scorer 1 with one false positive (which, however, was tagged as a skipped cycle by Scorer 2), and in 58 out of 64 (91%) cases detected by Scorer 2 with no false positives.

## Age and fractal cycles

Next, we tested whether fractal cycle duration changes as a function of age. We found that in the merged adult dataset (Datasets 1–5, n=205), the mean duration of the fractal cycles negatively correlated with the age of the participants ($r$=–0.19, $p$=0.006, age range: 18–75 years, median: 33.5 years, *Figure 1—figure supplement 4A*). Intriguingly, this correlation looked like a mirror image of the correlation between the age and wakefulness after sleep onset (*Figure 1—figure supplement 4B*). Following this observation, we performed an additional correlation between the fractal cycle duration and wakefulness proportion and found that it was non-significant ($r$=0.01, $p$=0.969). Nevertheless, we further performed a partial correlation between the fractal cycle duration and participant age, while controlling for the effect of wakefulness after the sleep onset and found that the correlation remained significant ($r$=–0.18, $p$=0.011).

Given that participant's age also correlated with REM latency (*Figure 1—figure supplement 4D*) while REM latency further correlated with fractal cycle duration (*Figure 1—figure supplement 4C*), we performed an additional partial correlation between the fractal cycle duration and age while controlling for REM latency. We found that it remained significant ($r$=–0.16, $p$=0.025). The partial correlation between the fractal cycle duration and REM latency adjusted for the participant's age was non-significant ($r$=0, $p$=0.746).

Of note, these correlations were significant while analyzing the pooled dataset only, they were not observed while analyzing each dataset separately. Moreover, when we added to the pooled adult dataset (Datasets 1–5) our pediatric dataset (Dataset 6), the correlation between fractal cycle duration and age became non-significant.

Interestingly, the mean duration of the classical cycles did not correlate with the age of the adult participants neither in the merged dataset (*r*=–0.02, p=0.751) nor while analyzing each dataset separately.

## Fractal cycles in MDD

Finally, to assess the clinical relevance of the fractal cycles, we explored them in patients with MDD. We found that patients at 7- and 28 day of medication treatment as well as long-termed medicated patients (Datasets A – C) showed a longer fractal cycle duration compared to controls with medium effect size (*Table 3*, *Figure 3B*). Moreover, in Dataset B, the patients who took REM-suppressive antidepressants (See *Appendix 1—table 5* for information on specific medications taken by the patients) showed longer fractal cycle duration compared to patients who took REM-non-suppressive antidepressants with medium effect size (70 cycles of 21 patients vs 63 cycles of 17 patients). In Dataset C, no difference was detected between these sub-groups. However, it should be noted that in Datasets C, the REM-suppressive and REM-non-suppressive antidepressant groups were unbalanced (87 cycles of 23 patients vs 35 cycles of 10 patients) and consisted of different medications than Dataset B.

*Table 3* and *Figure 3* show results calculated over frontal electrodes (or central ones for Dataset A). The topographical analysis over other areas is reported in *Appendix 1—table 6*.

In Dataset B (the only dataset including unmedicated patients), 7-day medicated patients had longer fractal cycles compared to their own unmedicated state with medium effect size (p=0.001, Cohen's d=0.4, *Figure 3A, B*, two additional examples are shown in *Figure 3—figure supplement 1*). Unmedicated patients and controls showed comparable durations of the fractal cycles. The only difference observed between these groups was a smaller amplitude of the fractal descent of the first fractal cycles in unmedicated patients compared to controls with a medium effect size (−3.2 to –3.6 z, p=0.040, Cohen's d=0.5).

In a pooled dataset, medicated patients showed a prolonged duration of fractal cycles compared to the controls (104±49 vs 88±31 min, p<0.001, *Figure 3C*). The between-group difference was the largest for the first cycle (*Figure 3D*). Moreover, cycle-to-cycle overnight dynamics of the fractal cycle duration showed a gradual decrease in medicated patients vs an inverted U shape in controls (*Figure 3D*).

To test our hypothesis that fractal cycles are more sensitive than classical cycles in detecting differences between patients and controls, we performed the same analysis as described above while using the duration of classical cycles as the variable of interest. The results were similar to those obtained for fractal cycle durations (*Table 3*, *Figure 3C, D*), that is our hypothesis was not confirmed. The comparable outcomes of the two analyses can be explained by the positive correlations between the durations of fractal and classical cycles observed in all groups of the medicated MDD patients like that seen in healthy controls (*Table 3*).

## Discussion

This study introduced the new concept of fractal activity-based cycles of sleep or 'fractal cycles' for short, which is based on temporal fluctuations of the fractal (aperiodic) slopes across a night. We showed that durations of these fractal cycles correlated with those of classical (non-REM – REM) sleep cycles defined by hypnograms in five independently collected datasets counting 205 healthy participants overall as well as in 111 medicated patients with MDD. Overnight cycle-to-cycle dynamics in healthy adults showed an inverted U-shape for both fractal and classical cycle durations. The fractal cycle algorithm was effective in detecting cycles with skipped REM sleep. The findings further revealed that children and adolescents showed shorter fractal cycles as compared to young healthy adults. In adults, fractal cycle durations negatively correlated with participants' age. Medicated patients with MDD showed longer fractal cycles compared to their own unmedicated state and healthy controls. Below we discuss these findings in detail.

### Fractal cycles: definition and motivation

We observed that the time series of fractal slopes have a cyclical nature, descending and ascending for about 4–6 times per night with a mean duration of approximately 90 min for each such ('fractal') cycle. This strikingly resembles the description of classical sleep cycles. Indeed, both the visual inspection

**Table 3.** Fractal cycles in MDD.

| Dataset | Group | No. participants | Age | Classical cycles | | | | Fractal cycles | | | | Fractal-classical cycles correlation | |
|---|---|---|---|---|---|---|---|---|---|---|---|---|---|
| | | | | No. cycles | Duration, min | p | d | No. cycles | Duration, min | p | d | r | p |
| A | HC (Dataset 1) | 38 | 46.8±10.7 | 171 | 86±23 | --- | --- | 167 | 84±35 | --- | --- | 0.33 | 0.042 |
| | long-termed med. MDD | 40 | 50.1±8.6 | 141 | 109±55 | $10^{-6}$ | 0.6 | 143 | 97±43 | 0.001 | 0.3 | 0.51 | 0.001 |
| B | HC (Dataset 2) | 39 | 31.0±9.9 | 180 | 90±21 | --- | --- | 171 | 90±26 | --- | --- | 0.51 | 0.001 |
| | unmed. MDD | 38 | 31.3±10.2 | 169 | 92±31 | n.s. | --- | 155 | 92±38 | n.s. | --- | 0.19 | n.s. |
| | 7d med. MDD | --- | --- | 149 | 102±43 | 0.003 | 0.4 | 133 | 107±51 | $10^{-4}$ | 0.5 | 0.68 | $10^{-6}$ |
| | REM-non-suppressive | 17 | 31.6±10.4 | 77 | 91±26 | --- | --- | 63 | 95±44 | --- | --- | 0.49 | 0.046 |
| | REM-suppressive | 21 | 33.6±11.3 | 72 | 103±54 | 0.002* | 0.5* | 70 | 121±55 | 0.003* | 0.5* | 0.66 | 0.001 |
| C | HC (Dataset 3) | 32 | 45.3±15.9 | 146 | 89±23 | --- | --- | 154 | 88±32 | --- | --- | 0.57 | 0.001 |
| | 7d med. MDD | 33 | 46.2±16.2 | 121 | 114±45 | $10^{-7}$ | 0.7 | 122 | 107±48 | $10^{-4}$ | 0.5 | 0.47 | 0.006 |
| | 28d med. MDD | --- | --- | 117 | 111±51 | $10^{-5}$ | 0.6 | 100 | 106±51 | 0.001 | 0.4 | 0.42 | 0.018 |

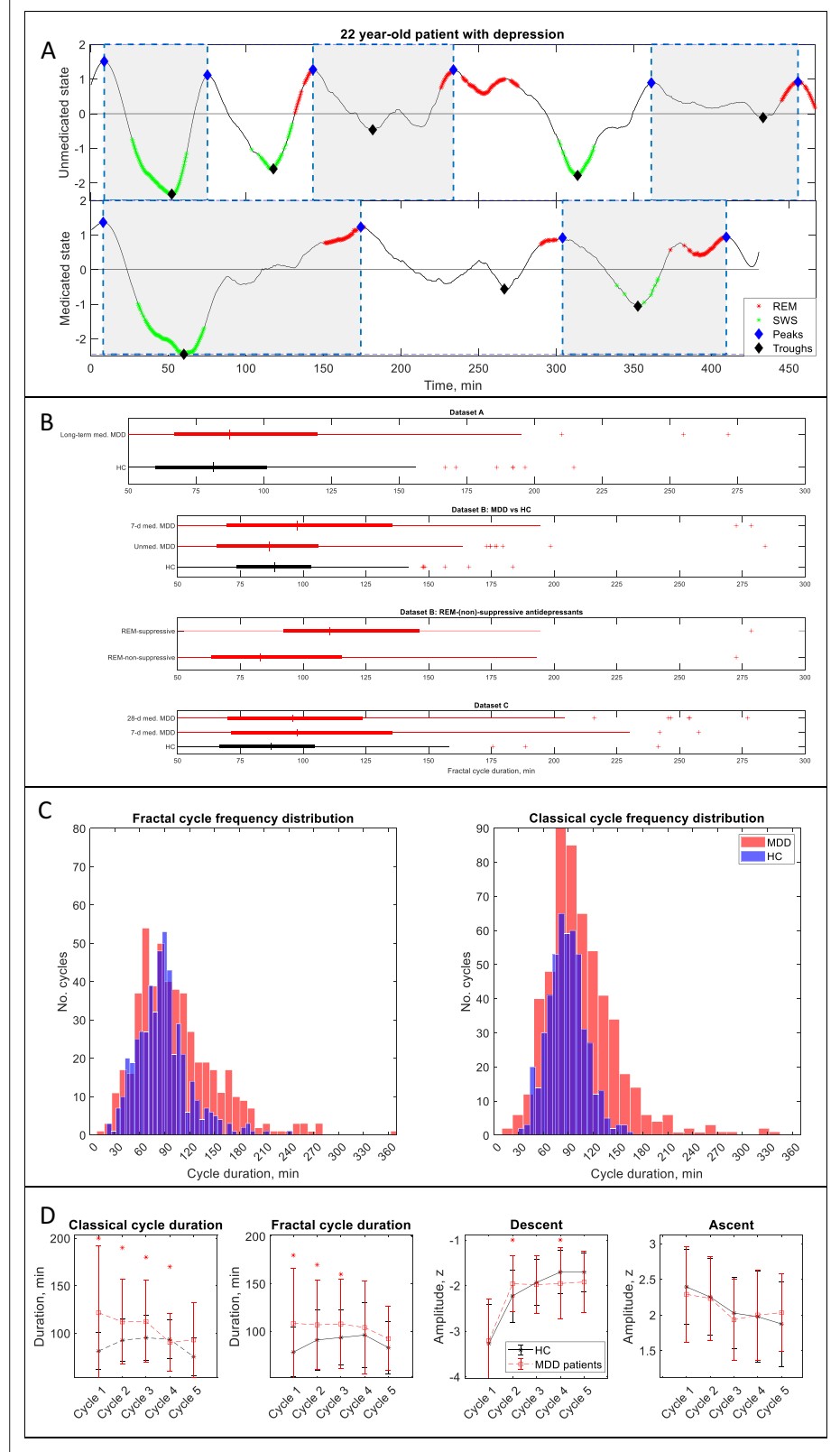

**Figure 3.** Fractal cycles in MDD. (**A**) Individual fractal cycles: time series of smoothed z-normalized fractal slopes observed in a 22 y.o. MDD patient (Dataset B) in their unmedicated (top) and 7-day medicated (bottom) states. Peaks (blue diamonds) are defined with the Matlab function *findpeaks* with the minimum peak distance of 20 min and minimum peak prominence of 0.9 z. Fractal cycles duration (defined as an interval of time between two

*Figure 3 continued*

successive peaks) is longer in the medicated compared to unmedicated states, reflecting shallower fluctuations of fractal (aperiodic) activity. Two additional patients are shown in *Figure 3—figure supplement 1*. (**B**) Box plots: the fractal cycle duration is longer in medicated MDD patients (red) compared to age and gender-matched healthy controls (black) in all datasets. In Dataset B, fractal cycles are longer in the medicated vs patients' own unmedicated state and in patients who took REM-suppressive vs REM-non-suppressive antidepressants. A vertical central line represents the median in each box, the left and right edges of the box indicate the 25th and 75th percentiles, respectively, the whiskers extend to the most extreme data points not considered outliers, and a plus sign represents outliers (individual cycles). (**C**) Frequency distribution: individual fractal and classical cycles pooled from three MDD datasets (**A – C**) are counted separately for medicated MDD patients and HC. (**D**) Overnight dynamics: cycle-to-cycle dynamics of the duration of both fractal and classical cycles show a gradual decrease in medicated patients vs an inverted U shape in controls. The between-group difference in cycle duration is the largest for the first cycle. Patients show flatter fractal descents of the second cycle and steeper fractal descents of the fourth cycle compared to controls. Contrary to controls, patients do not show a gradual decrease in absolute amplitudes of the fractal descents from the second to the fourth cycles. Patients and controls show comparable cycle-to-cycle dynamics of fractal ascents, * marks a statistically significant difference between the groups. MDD – major depressive disorder, HC – healthy controls, unmed. – unmedicated, med. – medicated, SWS – slow-wave sleep, REM – rapid eye movement.

The online version of this article includes the following figure supplement(s) for figure 3:

**Figure supplement 1.** Individual fractal cycles in MDD patients.

**Figure supplement 2.** Fractal cycles in patients with insomnia.

and formal correlational analyses revealed that the timing and duration of the fractal and classical cycles mainly matched. This led us to propose that *the 'fractal cycles of sleep' could serve as a new data-driven definition of sleep cycles*, that is a means to appreciate quantitatively what has been previously observed only qualitatively using hypnograms. Notably, we do not claim that fractal cycles are a substitute for the study of the individual sleep stages or microstructural features of sleep. We want to stress, however, that currently, sleep research is shifted towards the study of, to use a metaphor, 'the atoms' of sleep, such as individual sleep stages, slow oscillations, spindles, microarousals etc. Yet it is possible that some important (currently unknown) features of sleep could be explored only at the level of sleep cycles, 'the molecules of sleep'. (Note, that we use the molecule and atom concepts only as a metaphor for the macro- and microstructure of sleep.)

## Hypothetical functional significance of aperiodic activity and fractal cycles

The decision to incorporate fractal activity analysis in sleep cycle research was based on the reports that fractal (aperiodic) dynamics may reflect the bistability of the network (the overall tendency of alternating up and down states; *Baranauskas et al., 2012*) and/or alterations in the balance between neural excitatory and inhibitory currents (*Gao et al., 2017*). Circumstantial evidence suggests that fractal activity is a measure of sleep homeostasis or sleep intensity, reflecting sleep-wake history, sleep stage differences, sleep cycles, age-effects, local sleep and sleep disorders (*Bódizs et al., 2024*). Recently, it has been reported that during human sleep, spectral slopes positively correlate with pupil size, a marker of arousal levels linked to the activity of the locus coeruleus-noradrenergic system (*Carro-Domínguez et al., 2023*).

According to the reciprocal-interaction model of sleep cycles, each sleep phase is characterized by a specific neurochemical mixture. During non-REM sleep, aminergic inhibition decreases and cholinergic excitation increases such that at REM sleep onset, aminergic inhibition is shut off and cholinergic excitability reaches its maximum, while other outputs are inhibited (*Pace-Schott and Hobson, 2002*). Complex inhibitory and excitatory connections between pontine REM-on and REM-off neurons are further modulated by such neurotransmitters as GABA, glutamate, nitric oxide, and histamine. Intriguingly, during REM sleep, acetylcholine plays the main role in maintaining brain activation, which is expressed as EEG desynchronization, one of the main features of REM sleep, and other systems are silent (*Nir and Tononi, 2010*). This suggests that acetylcholine, which fluctuates cyclically across a night as a result of the REM-off – REM-on interactions, might have a key role in the sleep phase alternation.

**Table 4.** Hypothetical functional significance of fractal cycles.

| Theory/model | Reference | Hypothetical integration of the fractal cycle concept to the existing model |
|---|---|---|
| Two antagonistic roles of sleep:<br>1. sensory disconnection that facilitates restorative properties of sleep;<br>2. monitoring of the environment that transiently restores alertness. | *Simor et al., 2022* | • troughs of fractal cycles reflect (1);<br>• peaks of fractal cycles reflect (2). |
| Reactive and predictive homeostatic functions of sleep:<br>1. intensive restorative processes during early-night sleep;<br>2. active future-oriented processes during late-night sleep. | *Simor et al., 2023* | • deeper fractal cycles observed during early-night sleep reflect (1);<br>• shallower fractal cycles seen during late-night sleep reflect (2). |
| Reciprocal-interaction model of sleep cycles:<br>- alternations between non-REM and REM sleep stages are explained by the interaction between aminergic and cholinergic neurons of the mesopontine junction. | *Pace-Schott and Hobson, 2002* | • ascents and peaks of fractal cycles reflect acetylcholine release*;<br>• descents and troughs of fractal cycles coincide with aminergic activity. |
| Noradrenergic neurons create a non-reducible timeframe for the NREM-REM sleep cycle where low noradrenaline levels allow entries into REM sleep. | *Osorio-Forero et al., 2023*. | Ascents and peaks of fractal cycles reflect a cease of noradrenaline release. |
| The Neuronal Transition Probability Model:<br>1. During a move towards deep sleep beta power drops exponentially, delta power rises in an S-curve and sigma power peaks while delta is still rising;<br>2. During a move away from deep sleep, delta drops, beta rises. | *Merica and Fortune, 2011* | • descending part of the fractal cycle corresponds to (1);<br>• ascending part of the fractal cycle corresponds to (2). |

*this hypothesis is also based on the report that in rats, cholinergic nucleus basalis stimulation caused flattering of spectral decay (*Goard and Dan, 2009*).

Given that the specific neurochemical milieu of the brain produces a specific type of conscious experience (*Nir and Tononi, 2010*) and that conscious experience was shown to be related to fractal activity derived from the human sleep EEG (*Colombo et al., 2019*), it is tempting to speculate that fractal activity tracks sleep-related changes in the neurochemical milieu of the brain and overall network dynamics. This has not been tested in humans; nevertheless, in rats, cholinergic nucleus basalis stimulation acutely increased higher to lower frequency cortical LFP power ratio or in other words, caused flattering of spectral decay (*Goard and Dan, 2009*). One can, therefore, speculate that ascending parts and peaks of fractal cycles coincide with acetylcholine release. The troughs of fractal cycles, in turn, might reflect a higher homeostatic pressure and even cause feelings of sleepiness and the search for the opportunity of initiating sleep, as these are periods of the steepest fractal activity, which implies a higher ratio of lower over higher frequency power in the EEG (*Bódizs et al., 2024*).

In view of this literature, we speculate that fractal fluctuations may reflect two antagonistic roles of sleep (*Simor et al., 2022*). Specifically, fractal cycle troughs might cohere with sensory disconnection that facilitates restorative properties of sleep while fractal cycle peaks reflect monitoring of the environment that transiently restores alertness (*Table 4*).

## Fractal and classical cycles comparison (Table 5)

In this study, in healthy adults, 81% of all fractal cycles defined by our algorithm could be matched to individual classical cycles defined by hypnograms. Correlations between the durations of fractal and classical cycles were observed not only in healthy adults but also in MDD patients who took antidepressants. The results show that displaying sleep data using fractal activity as a function of time meaningfully adds to the conventionally used hypnograms thanks to the gradual and objective quality of fractal power.

Thus, in hypnograms, each sleep stage is ascribed with a categorical value (e.g. wake = 0, REM = –1, N1 = –2, N2 = –3 and SWS = –4, *Figure 1A*). Yet categorical labeling of sleep stages can induce information loss and lead to several misinterpretations, such as an implied order of sleep stages (e.g. 'REM sleep is located between wake and N1') and an implied 'attractor state' conception of sleep stages (e.g. 'no inter-stage states'). Likewise, defining the precise beginning and end of a classical sleep cycle using a hypnogram is often difficult and arbitrary, for example, in cycles with skipped or interrupted REM sleep or REM sleep without atonia.

In contrast, fractal cycles do not rely on the assignment of categories, being based on a real-valued metric with known neurophysiological functional significance. This introduces a biological

**Table 5.** Fractal and classical cycle comparison.

| Fractal cycles by our algorithm | Classical cycles by hypnograms |
|---|---|
| **Definition/detection** | |
| Based on a real-valued metric with known neurophysiological functional significance | Based on categorical values of the cycle constituents (e.g. wake = 0, REM = –1, N1 = –2, N2 = –3 and SWS = –4) |
| Gradual changes | Abrupt changes |
| Automatic computation, objective | Usually based on the visual inspection, time-consuming, subjective, error-prone |
| **Findings** | |
| Cycles with skipped REM sleep detected in 91–98% of cases | Inter-rater agreement of 91% on the presence of cycles with skipped REM sleep |
| Fractal cycle durations negatively correlated with the age of adult participants | Classical cycle durations did not correlate with the age of adult participants |
| Shorter fractal cycle durations in children vs adults: higher F-values, R², effect sizes than for classical cycles | Shorter classical cycle durations in children vs adults: lower F-values, R², effect sizes than for fractal cycles |
| Shorter first and second fractal cycles in the pediatric group | No difference in durations of the first and second classical cycles in pediatric vs adult groups |
| No difference in duration of the fourth fractal cycles in the pediatric group | Shorter duration of the fourth classical cycle in the pediatric group |
| Longer fractal cycle duration in medicated patients with depression: comparable differences with those on classical cycles | Longer classical cycle duration in medicated patients with depression: comparable differences with those on fractal cycles |

**Sources of mismatches between fractal and classical cycles**

| Source | Finding | Reason |
|---|---|---|
| Across night variation in *REM sleep episode duration:* longer REM episodes towards morning | Longer REM episodes are associated with a higher mismatch between fractal vs classical cycles | The end of a fractal cycle is defined as the local maximum of time series of fractal slopes, whereas the end of a classical cycle is defined as the end of the REM episodes |
| Across subject variation in *WASO:* a higher WASO proportion in older participants | A higher WASO proportion is associated with a higher mismatch between fractal vs classical cycles | REM- and wake-related smoothed fractal slopes show close values, therefore, both could be defined as local peaks. More fractal peaks imply more fractal cycles |

REM – rapid eye movement, SWS – slow-wave sleep, WASO – wake after sleep onset.

foundation and a more gradual impression of nocturnal changes compared to the abrupt changes that are inherent to hypnograms.

Importantly, fractal cycle computation is automatic and thus objective. Even though recently, there has been a significant surge in sleep analysis incorporating various machine learning techniques and deep neural network architectures, we should stress that this research line mainly focused on the automatic classification of sleep stages and disorders almost ignoring the area of sleep cycles. Here, as the reference method, we used one of the very few available algorithms for sleep cycle detection (*Blume and Cajochen, 2021*). We found that automatically identified classical sleep cycles only moderately correlated with those detected by human raters (r's=0.3–0.7 in different datasets). These coefficients lay within the range of the coefficients between fractal and classical cycle durations (*r*=0.41–0.55, moderate) and outside the range of the coefficients between classical cycle durations detected by two human scorers (r's=0.7–0.9, strong, *Appendix 1—table 8*).

One of the most significant methodological strengths of the fractal cycle algorithm is its ability to detect cycles with skipped REM sleep common in children, adolescents and young adults. Our algorithm detected skipped cycles in 91–98% of cases. We deduce that the fractal cycle algorithm detected skipped cycles since a lightening of sleep that replaces a REM episode in skipped cycles is often expressed as a local peak in fractal slope time series. Based on this, we further hypothesize that,

analogously, fractal cycles might detect REM sleep without atonia episodes in REM sleep behaviour disorder, the episodes currently often mistaken as non-REM sleep.

The mismatches between fractal and classical cycle numbers and durations (observed in 19% of cases) were mainly related to longer WASO or REM sleep episode durations (*Table 5*). The latter finding is in line with our recent study that has shown that fractal activity within a REM sleep episode is not homogenous, with phasic states showing steeper fractal slopes than tonic ones (*Rosenblum et al., 2025*). Future research should define whether the mismatches between fractal and classical cycles (when present) are the disadvantages of our algorithm or, on the contrary, the reflection of its ability to measure the cycling nature of sleep in a more precise way than the classical cycles.

In summary, we expect that fractal cycles could bring insights into (yet) unexplained phenomena thanks to their gradual and objective quality, and, therefore, have the potential to induce a paradigm shift in basic and clinical (see below) sleep research.

## Fractal slopes and SWA: overnight dynamics

Of note, currently, the gold standard marker of many sleep functions (e.g. restorative, regenerative) with a long-standing use is slow-wave activity (SWA), which, similar to fractal slopes, is also continuous and objective. SWA, however, has several disadvantages, such as large variability between individuals, which makes it impossible to set up a given reference point for healthy sleep (*Horváth et al., 2022*). Interindividual variability of spectral slopes is much smaller compared to SWA, making it a less individual-specific metric, yet spectral slopes strongly correlate with SWA (31–53% of shared variance throughout the non-REM periods; *Horváth et al., 2022*; *Bódizs et al., 2024*). In addition, both the literature and our findings show that while SWA has a cycling nature during the first part of the night, neural dynamics of late-night's sleep are not reflected by SWA at all (*Figure 4—figure supplement 9*). Given that SWA is a primary marker of sleep homeostasis, this pattern possibly reflects the dissipation of a sleep need over the night (*Bódizs et al., 2024*). In contrast, fractal slopes show a cycling nature over the entire night's sleep (*Figure 1A, B*), suggesting that they are a more suitable means to reflect the macrostructure of the whole night's sleep than SWA.

Having said this, we should highlight that characteristics of fractal cycles of sleep do undergo some overnight changes. Thus, the durations of both fractal and classical cycles in health show an inverted U-shape across a night and the amplitudes of fractal descents and ascents are larger during early-night- compared to late-night cycles (*Figure 1D*). This is in line with the report on the flattening of fractal activity from early to late sleep cycles (*Horváth et al., 2022*). If seen in the context of the reactive and predictive homeostatic functions of sleep (*Simor et al., 2023*), deeper fractal cycles observed during early-night sleep could reflect intensive restorative processes (which are also reflected by SWA), whereas shallower fractal cycles seen during the later part of night's sleep could reflect more active future-oriented processes (which are not reflected by SWA) with a shift towards neural excitation relative to inhibition expressed as overall flatter fractal activity (*Table 4*).

## Fractal cycles and age

We found that older healthy participants had shorter fractal cycles compared to the younger ones while classical cycles did not correlate with the participants' age. At first glance, it looked as if this association simply reflected an increased proportion of the wake after the sleep onset often seen in older adults (*Figure 1—figure supplement 4B*). Indeed, our algorithm does not discriminate between the smoothened wake- and REM-related fractal slopes and can define both as local peaks (*Figure 1A, B*). This happens because for the most part, wake- and REM sleep-related smoothed fractal slopes display comparable values, which are also the highest ones compared to other stages (*Figure 4—figure supplement 4*, green squares). Since the fractal cycle duration is defined as an interval of time between two adjacent peaks, more awakenings during sleep are expected to result in more peaks and, consequently, more fractal cycles per total sleep time, that is a shorter cycle duration. It is worth mentioning that unsmoothed wake- and REM-related slopes differ *Schneider et al., 2022* and *Figure 4—figure supplement 4* here (black squares). However, this is a side notion as raw values were not used in this study since our algorithm performed poorly on raw time series.

Moreover, a larger difference in classical vs fractal cycle duration was associated with a higher proportion of wake after sleep onset (WASO) in 3/5 datasets as well as in the merged dataset (*Table 2*). On the other hand, the partial correlation between fractal cycle duration and age remained significant

after controlling for the WASO amount. This hints that the association between fractal cycles and age might reflect more than just a confounding effect of WASO. This interpretation is in line with literature on age-related changes in aperiodic activity, namely, on flattering of fractal slopes with age (*Voytek and Knight, 2015*; *Bódizs et al., 2021*; *Pathania et al., 2022*), especially during SWS (*Schneider et al., 2022*). Likewise, aging is associated with shorter and fewer classical cycles, with a mean of 3.5 cycles per night compared to the usual 4–5 in adults and adolescents (*Conte et al., 2014*). Our findings suggest that fractal cycles are more sensitive to these age-related alterations than the classical ones. We further speculate that the claim that 'age affects sleep microstructure more than sleep macrostructure' (*Schwarz et al., 2017*) might reflect the lack of a reliable measure of sleep cycles.

Another plausible explanation for longer fractal cycles in younger compared to older adults could be rooted in increased sleep intensity of the younger adults (*Jenni and Carskadon, 2004*). Further, high sleep intensity driven by homeostatic pressure is associated with the delay in the emergence of the REM sleep phase (*Le Bon, 2020*; *Tarokh et al., 2012*). In our dataset, REM latency also decreased with age. Thus, *Figure 1—figure supplement 4D* illustrates that young adults might present with very delayed REM latency, that is 200–250 min after sleep onset, in line with the notion that younger adults more often show cycles with skipped REM sleep (*Figure 1—figure supplement 3*). This can be partly explained by the fact that younger people often have a later chronotype ('night owls') than older people with puberty linked to delays in the sleep cycle by up to 2 hr (*Randler, 2016*). Young people also have a longer circadian rhythm (>24 hr) than older ones (<24 hr, *Monk, 2005*).

To further strengthen this line of explanations, we performed a supplemental analysis, which showed that prolonged REM latencies are indeed associated with longer fractal cycles (*Figure 1—figure supplement 4C*). Nevertheless, the correlation was weak (yet significant) and observed in the pooled dataset only, that is not while analyzing individual datasets. Likewise, the partial correlation between the fractal cycle duration and REM latency adjusted for the participants' age was non-significant. Moreover, we found that children and adolescents (the group that has the longest REM latencies and the highest rate of cycles with skipped REM sleep) showed shorter fractal cycles compared to young adults, specifically the early-night fractal cycles. In view of these analyses, our attempt to explain longer fractal cycles in younger compared to older adults by increased REM sleep latency becomes less convincing. Moreover, given that our algorithm does not miss cycles with skipped REM sleep, longer REM sleep latencies should not necessarily be related to longer cycles. To summarize, at this stage, the mechanism underlying age-related differences in fractal cycle duration is unclear (possibly with some non-linearities) and future studies are needed to corroborate and further explore it.

## Fractal cycles in MDD

In addition, our study shows that deviations from the observed fractal patterns have some clinical relevance. We found that MDD patients in the medicated state had longer fractal cycles compared to their own unmedicated state and healthy controls. The largest differences were observed for the first sleep cycles. Moreover, patients who took REM-suppressive antidepressants showed prolonged fractal cycles compared to patients who took REM-non-suppressive antidepressants. Given that the fractal cycle duration was defined as an interval of time between two adjacent peaks and that the peaks usually coincide with REM sleep (*Figure 1A*), this finding may reflect such aftereffects of antidepressants as delayed onset and reduced amount of REM sleep (*Palagini et al., 2013*). In other words, if a patient has fewer REM sleep episodes, then the time series of their fractal slopes has fewer peaks and the algorithm detects fewer cycles per total sleep time, that is cycle's duration is longer (*Figure 3A*).

Another explanation considers our previous finding that medicated MDD patients show flatter average fractal slopes compared to controls and their own unmedicated state during all sleep stages (*Rosenblum et al., 2023a*). This might mean that the antidepressant intake results in shallower fractal fluctuations, which in turn implies that fewer peaks could be detected by our algorithm as the peak threshold was defined a priori in a healthy – not MDD – sample. Interestingly, recently, flatter fractal slopes during REM sleep have been also associated with sustained polyphasic sleep restriction in health (*Rosenblum et al., 2024b*), whereas flatter fractal slopes during non-REM sleep were observed in patients with objective insomnia and sleep state misperception, reflecting an abnormally high level of excitation in line with the hyperarousal model of insomnia (*Andrillon et al., 2020*). Our pilot findings

have shown that patient with psychophysiological insomnia have shorter fractal cycles compared to controls (*Figure 3—figure supplement 2*).

## Limitations and strengths

The major limitation of this study is its correlational approach, and thus an inability to shed light on the mechanism underlying sleep cycle generation. Therefore, the question of what determines the number and duration of cycles per night remains open. Moreover, further work is needed to determine the mathematically precise and physiologically meaningful model of fractal cycles. Notably, here, we suggest that fractal cycles are a new tool to study the macrostructure of sleep; however, they are presumably not a substitute for the study of the individual sleep stages and microstructural features of sleep (e.g. microarousals, spindles, slow waves).

Additionally, we explored the effect of developmental changes and aging on fractal cycles using a cross-sectional observational approach, whereas these factors might be disentangled more precisely in a longitudinal approach. The age of the pediatric group ranged from 8 to 17 years old; studying younger children and babies would add crucial information on the influence of neurodevelopmental changes on fractal cycles.

The strengths of this study are its large sample size, scripts and data sharing and self-replications in several clinical and healthy datasets of participants in a broad age range, affirming the overall robustness of the phenomena of fractal cycles. Another strength of this work is its generalizability as it has shown that the studies conducted in different experimental environments (including one study conducted at home) using different EEG devices provide comparable results.

To summarize, the large sample and self-replication performed in this study suggest that the 'fractal cycle' is a universal concept that should be extensively studied. Displaying the data in the format of fractal cycles provides an intuitive and biologically plausible way to present whole-night sleep neural activity and also adds some graduality to the purely categorical concept of sleep stages that comprise a hypnogram. In future studies, this graduality might help to illuminate differences in sleep architecture across different species, advance our understanding of the role of sleep in neurocognitive development in infants and adolescents as well as in neurodegenerative processes and other fields of neuroscience.

## Conclusion

We observed that the slopes of the fractal (aperiodic) spectral power descend and ascend cyclically across a night such that the peaks of the time series of the fractal slopes coincide with REM sleep or sleep lightening while the troughs of these time series coincide with non-REM sleep. Based on this observation, we introduced a new concept of fractal activity-based cycles of sleep or 'fractal cycles' for short, defining it as a time interval between two adjacent local peaks of the fractal time series. We have shown that fractal cycles defined by our algorithm largely coincide with classical (non-REM – REM) sleep cycles defined by a hypnogram and replicated our findings in several independently collected healthy and clinical datasets. Moreover, we found that the fractal cycle algorithm reliably detected cycles with skipped REM sleep. In addition, we observed that fractal cycle duration changes as a non-linear function of age, being shorter in children and adolescents compared to young adults as well as in older compared to younger adults. To this end, we conclude that the fractal cycle is an objective, quantifiable and universal concept that could be used to define sleep cycles and display the whole-night sleep neural activity in a more intuitive and biologically plausible way as compared to the conventionally used hypnograms. Having shown that the fractal cycles are prolonged in medicated patients with MDD, we suggest that fractal cycles are a useful tool to study the effects of antidepressants on sleep. Possibly, fractal cycles also will be able to serve as a means to explore sleep architecture alterations in different clinical populations (e.g. to detect REM sleep without atonia) and during neurocognitive development. In summary, this study shows that the fractal cycles of sleep are a promising research tool relevant in health and disease that should be extensively studied.

## Materials and methods
### Healthy participants

We retrospectively analyzed polysomnographic recordings from the following studies (*Table 6*):

**Table 6.** Datasets description.

| Characteristic | Dataset 1 (A) | Dataset 2 (B) | Dataset 3 (C) | Dataset 4 | Dataset 5 | Dataset 6 (pediatric) |
|---|---|---|---|---|---|---|
| Reference to original study | *Rosenblum et al., 2023a* | | | *Jafarzadeh Esfahani et al., 2023* | *Rosenblum et al., 2024a* | *Furrer et al., 2019; Volk et al., 2019; Jaramillo et al., 2020* |
| No. healthy participants (-excluded) | 40 (-2) | 40 (-1) | 33 (-1) | 36 (-2) | 68 (-6) | 21 (0) |
| Exclusion reasons | >25% WASO <150 min recording | <150 min recording | >25% WASO | >25% WASO No REM | >25% WASO No REM | — |
| No. MDD patients (none excluded) | 40 | 38 | 33 | 0 | 0 | 0 |
| Study environment | Sleep lab +a memory task before* | Sleep lab +memory tasks before*,† | Sleep lab | Sleep at home with EEG and headband | Sleep lab +simultaneous blood measurement ‡ | Sleep lab +MRI before and after sleep § |
| Device | Comlab 32 Digital Sleep Lab, Brainlab V 3.3 Software, Schwarzer, GmbH, Munich, Germany | JE-209A amplifier (Nihon Kohden, Tokyo, Japan), with 128ch BrainCap (EasyCap GmbH, Herrsching, Germany) | Comlab 32 Digital Sleep Lab, Brainlab V 3.3 Software, Schwarzer GmbH, Munich, Germany | Somnomedics GmbH, Randersacker, Germany | Comlab 32 Digital Sleep Lab, Brainlab V 3.3 Software, Schwarzer GmbH, Munich, Germany | Sensor Net for long-term monitoring (Electrical Geodesic Inc, EGI, Eugene, OR, USA) |
| No. channels | 4 | 128 | 32 | 24 | 16 | 128 |
| (Offline re)-referenced to | Contralateral mastoid | Average of all leads | Average of all leads | Contralateral mastoid | Contralateral mastoid | Contralateral mastoid |
| Sample rate, Hz | 250 | 200 | 250 | 256 | 250 | 500 |
| Filtering during recording, Hz | 0.3–70 | >0.016 | 0.53–70 | 0.2–35 | 0.3–70 | 0.01–200 |
| Available frontal electrodes | none | Fz, F1, F2, F3, F4, F5, F6, F7, F8, F9, F10 | Fz, F3, F4, F7, F8 | F3, F4 | F3, F4 | F3, F4 |
| Analyzed electrodes | C3, C4 | F3, F4 | F3, F4 | F3, F4 | F3, F4 | F3, F4 |

WASO – wake after sleep onset, REM – rapid eye movement sleep, MDD – major depressive disorder.

*a procedural memory paradigm (finger tapping task) before sleep.

†a declarative memory paradigm (word-pair learning task) before sleep.

‡in this study, 4 ml blood were drawn every 20 min from the adjacent room, using an intravenous cannula and a tube extension.

§an MRI scan was taken in the evening before and in the morning after the sleep measurement.

## Datasets 1–3

40, 40, and 33 healthy controls from three independent sleep studies in MDD conducted at the Max Planck Institute of Psychiatry, Germany. These datasets are described in *Rosenblum et al., 2023a* and *Bovy et al., 2022*. In addition, these participants are used as controls in MDD datasets A – C described below.

## Dataset 4

36 healthy participants from a home-based sleep study exploring simultaneous polysomnographic and EEG wearables conducted at the Donders Institute for Brain, Cognition and Behavior, the Netherlands (Described as Dataset 2 in *Jafarzadeh Esfahani et al., 2023*). The signal was recorded at participants' homes over three nights with a gap of a week between each recording. For consistency with other datasets (i.e. to end up with a comparable number of cycles provided by each participant), we used polysomnography (and not EEG recorded by wearables) from the first night only since it had the largest sample size (i.e. 5 subjects dropped out from the study after the first polysomnographic recording).

### Dataset 5

68 healthy controls from previous endocrinological studies conducted at the Max Planck Institute of Psychiatry, Germany, using only nights with no pharmacological or endocrine intervention. 60/68 participants are described in *Rosenblum et al., 2024a*.

### Dataset 6

21 healthy children and adolescents from previous studies (*Furrer et al., 2019*; *Volk et al., 2019*; *Jaramillo et al., 2020*) conducted at the University Children's Hospital Zürich, Switzerland. For the control group to this dataset, we selected all healthy adults from Datasets 1–3, 5, 6 (n=205) whose ages lay in the range of 23–25 years (the age when the brain maturation process is supposed to be finished *Giedd and Rapoport, 2010* and no age-related processes are expected to start). This resulted in 24 subjects with a mean age of 24.8±0.9 years.

The studies were approved by the Ethics committee of the University of Munich (Datasets 1–3, 5), Radboud University (Dataset 4) and Canton of Zürich (Dataset 6). All participants (or participants' parents for Dataset 6) gave written informed consent.

## Patients with MDD

We retrospectively analyzed polysomnographic recordings from our previous studies (*Bovy et al., 2022*; *Rosenblum et al., 2023a*, *Table 1*, *Table 2*):

### Dataset A

40 long-term medicated MDD patients vs 40 age- and gender-matched healthy controls (Dataset 1 here).

### Dataset B

38 MDD patients in unmedicated and 7-day medicated states vs 40 healthy age and gender-matched controls (Dataset 2 here).

### Dataset C

33 MDD patients at 7 day and 28 day of medication treatment vs 33 healthy age and gender-matched controls (Dataset 3 here).

Demographic and sleep characteristics of the patients, medication treatment and polysomnographic devices are described in our previous works (*Bovy et al., 2022*; *Rosenblum et al., 2023a*). Here, *Appendix 1—table 5* presents medication treatment. In *Rosenblum et al., 2023a*, Datasets A, B, and C are referred to as the Replication Dataset 2, Main Dataset and Replication Dataset 1, respectively; in *Bovy et al., 2022*, the naming is the same as here. All studies were approved by the Ethics committee of the University of Munich. All participants gave written informed consent.

The first part of this study analyzes the data from healthy participants only and labels the datasets with the numbers 1–6. The second part of this study compares patients and controls and labels the analyzed datasets with the letters A – C. Notably, healthy participants used as controls in datasets A – C are the same subjects analyzed in Datasets 1–3.

In Appendix, we report how many participants and for what reasons were excluded from the analysis. An example of one excluded participant is given in *Figure 1—figure supplement 3* C (S37). Likewise, we report pilot findings on fractal cycles in patients with psychophysiological insomnia, using the open access dataset from *Rezaei et al., 2017* (*Figure 3—figure supplement 2*).

## Polysomnography

Information about the studies and polysomnographic devices is reported in *Table 6*. The participants slept wearing a polysomnographic device in a sleep laboratory (Datasets 1–3, 5, 6) or in the home environment (Dataset 4). In datasets 1–3 and 5, all participants had an adaptation night before the examination night; adaptation night data was not available to be analyzed and reported here. In dataset 6, all participants had two recording nights: a baseline and an examination night with auditory stimulation. Here, only the baseline night was analyzed, which was either the first night (in 50% of cases) or the second night for a given participant.

Sleep stages were previously scored manually by independent experts according to the AASM standards (*American Academy of Sleep Medicine, 2014*). In the pediatric dataset, we used 20 s epochs, in the rest of the datasets, we used 30 s epochs. Epochs with EMG and EEG artifacts and channels with more than 20% artifacts during non-REM sleep were manually excluded by an experienced scorer before all automatic analyses.

We opted to analyze the F3 and F4 electrodes for maximal consistency between the studies as these leads were available in 6 out of 7 datasets. Another reason is that in our future studies, we plan to replicate this work using the data recorded with at-home wearable devices, which often have only frontal channels (e.g. F7 and F8). We report the topographical analysis over central, parietal and occipital electrodes (when available) in healthy and clinical datasets in *Appendix 1—table 1*; *Appendix 1—table 6* respectively, showing comparable results. In *Appendix 1—table 1*, we also report correlations between fractal cycle durations defined using different channels.

## Fractal power component

The analysis flowchart is depicted in *Figure 4—figure supplement 1*. Outputs of some of the analysis steps in an example individual are shown in *Figure 4*.

Offline EEG data analyses were carried out with MATLAB (version R2021b, The MathWorks, Inc, Natick, MA), using the Fieldtrip toolbox and custom-made scripts. For each participant, we averaged the EEG signal over the F3 and F4 electrodes (or C3 and C4 – for Dataset 1 where the frontal channels were unavailable), calculated its spectral power for every 30 (adult datasets) or 20 (the pediatric dataset) seconds corresponding to the conventionally defined duration of sleep epochs and differentiated the total power to its fractal (i.e. aperiodic, 1 /f, scale-free) and oscillatory components. Several methods to calculate fractal components exist. We opted to use the Irregularly Resampled Auto-Spectral Analysis (IRASA; *Wen and Liu, 2016*) tool embedded in the Fieldtrip toolbox (*Oostenveld et al., 2011*), one of the leading open-source EEG softwares, with the *ft_freqanalysis* function as described elsewhere (*Rosenblum et al., 2023b*). A side note: slopes calculated with the IRASA strongly correlate (r = |0.9|) with those calculated using the 'fitting oscillations and one over f' (FOOOF, *Schneider et al., 2022*), another useful method used for aperiodic analysis (*Donoghue et al., 2020*). The fractal power component (shown in *Figure 4—figure supplement 2*) was transformed to log-log coordinates and its slope was calculated to estimate the power-law exponent (the rate of spectral decay), using the function *logfit* (*Lansey, 2020*). The loglog data fit is shown in *Figure 4—figure supplement 3*.

As opposed to the oscillatory component, the fractal component is usually treated as a unity and, therefore, is filtered in the broadband frequency range (*Donoghue et al., 2020*; *Bódizs et al., 2021*; *Gerster et al., 2022*). Nevertheless, different studies defined (slightly) differing bands, for example 30–50 Hz (*Gao et al., 2017*; *Lendner et al., 2020*), 3–55 Hz (*Waschke et al., 2021*), 0.5–35 Hz (*Miskovic et al., 2019*), 1–40 Hz, 1–20 Hz and 20–40 Hz (*Colombo et al., 2019*), 1–45 Hz (*Helson et al., 2023*), 0.5–40 Hz (*Vinding et al., 2021*), 3–45 Hz and 30–45 Hz (*Höhn et al., 2024*) and 2–48 Hz (*Bódizs et al., 2021*; *Schneider et al., 2022*).

Here, we used the 0.3–30 Hz range as this is a typical sleep frequency band used in many areas of sleep research, showing good ability to differentiate between sleep stage as could be seen in *Figure 4—figure supplement 4*, which replicates existing literature. Dataset 4 was analyzed in the 0.3–18 Hz range since relatively low low-pass filtering was applied to it during the recording (see *Table 6*). In *Appendix 1—table 2*, we also analyze the 1–30 Hz band to control for a possible distortion (the so called "knees'' of the spectrum) of the linear fit by excluding low frequencies with strong oscillatory activity (*Gao et al., 2017*; *Bódizs et al., 2021*). We find that the results are similar to those obtained for the 0.3–30 Hz band reported in the Main text (probably thanks to the smoothening procedure we applied).

Finally, *Figure 4—figure supplement 5* shows aperiodic slopes in the 30–48 Hz band averaged over sleep stages for Datasets 1–3 and 5. According to literature, REM sleep is expected to show the steepest (most negative) high-band slopes compared to all other sleep stages. However, we were able to replicate this finding in Datasets 1 and 5 only. Given poor differentiation between the stages in 2/4 datasets, this variable was not used in any further analyses.

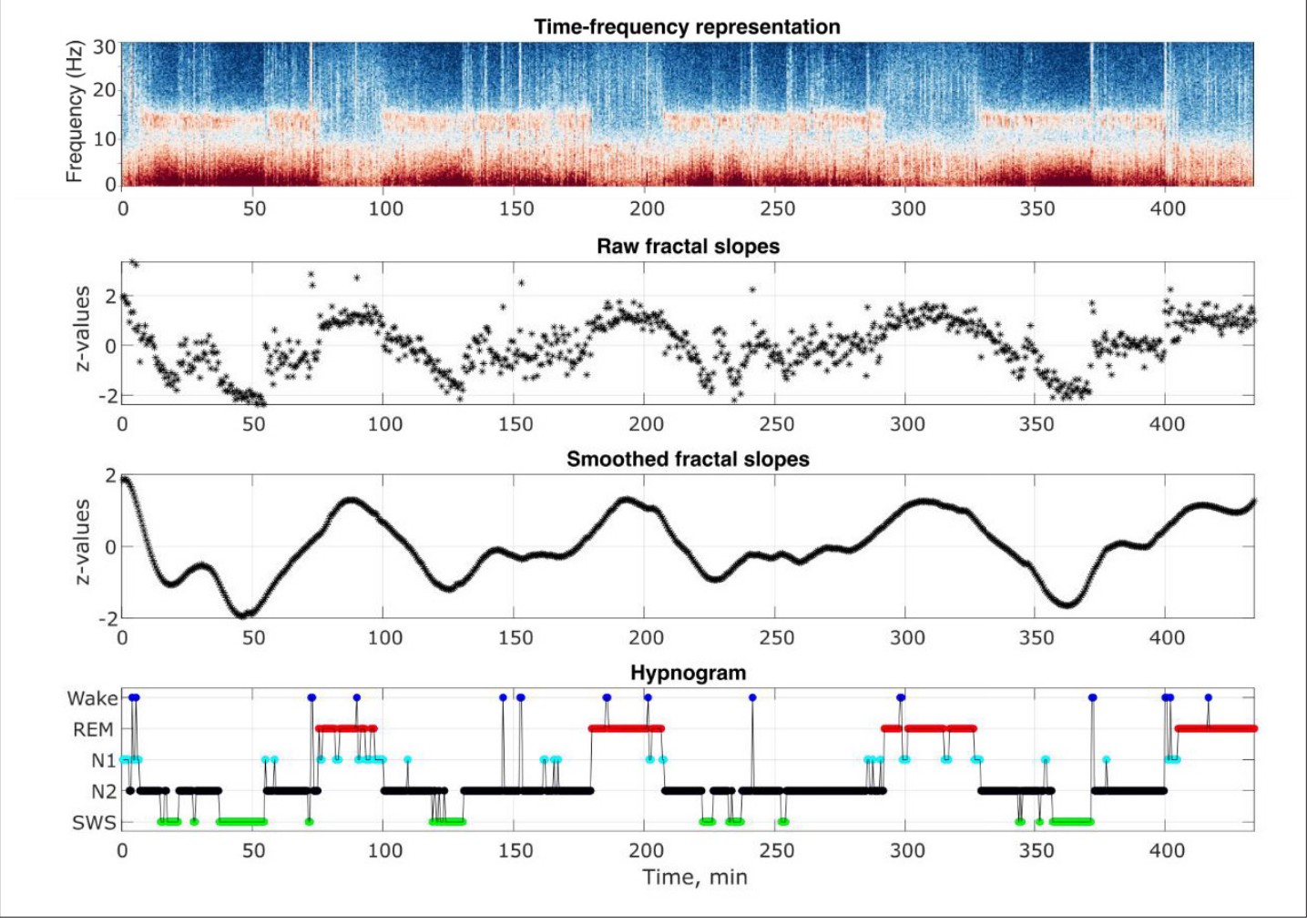

**Figure 4.** Analysis output examples. Outputs of some of the analysis steps in an example healthy 26-year-old individual. From top to bottom: time-frequency representation of the total spectral power, raw and smoothed time series of the fractal slopes and hypnogram. Frontal spectral power and its slopes were calculated in the 0.3–30 Hz range for each 30 s of sleep.

The online version of this article includes the following figure supplement(s) for figure 4:

**Figure supplement 1.** Analysis flowchart.

**Figure supplement 2.** Fractal power component.

**Figure supplement 3.** Log-log fit of data.

**Figure supplement 4.** 0.3–30 Hz z-normalized slopes.

**Figure supplement 5.** 30–48 Hz z-normalized slopes.

**Figure supplement 6.** 0.3–30 Hz raw slopes.

**Figure supplement 7.** Autocorrelation (*left*) and partial autocorrelation (*right*) of the time series of the fractal slopes averaged over each 30 s of sleep show the correlation of this time series with a delayed version of itself as a function of time lag.

**Figure supplement 8.** Cross-correlations.

**Figure supplement 9.** Individual fractal *vs.* SWA time series.

## Fractal activity-based cycles of sleep

Fractal activity-based cycles of sleep or 'fractal cycles' for short were defined from fractal slope time series. For this, time series of the fractal slopes were z-normalized (raw values can be seen in *Figure 4—figure supplement 6*) within a participant and smoothened with the Savitzky-Golay filter (*Figure 4*), the filter highly used in many fields of data processing. We used the Matlab's function *sgolayfilt(slope_time_series, order, frame_length)* with the polynomial order of five and the frame

length of 101. The peaks of the smoothed time series of the fractal slopes were defined with Matlab's function *findpeaks (slope_time_series, 'MinPeakDistance', 40, 'MinPeakProminence', 0.9)* with the minimum peak distance of 20 min (i.e. forty 30 s epochs) and minimum peak prominence of |0.9| z (*Figure 4*). The amplitude of the descending and ascending phases of a cycle was defined to be > |0.9| z, meaning that there is a probability of *P*=0.8 that a given fractal slope lies below/above the standard normal distribution.

Of note, we had no solid a priori theoretical indication for choosing either of the function settings mentioned above. All settings were chosen a posteriori following an exploratory visual inspection of the normalized data from one dataset (Dataset 5), which therefore can be transferred to other datasets. That is, in datasets 1–4 and 6, the settings of the *sgolayfilt* and *findpeaks* functions were defined a priori based on the results obtained while inspecting Dataset 5.

In *Appendix 1—table 7*, we compare results obtained while using different thresholds of the abovementioned parameters; namely, longer and shorter smoothing windows and higher and lower minimum peak prominence.

## Classical sleep cycles

Classical sleep cycles were defined manually via the visual inspection of the hypnograms by two independent scorers according to the criteria originally proposed by *Feinberg and Floyd, 1979* with some adaptations as follows. A cycle typically starts with N1, N2 or sometimes wake and is followed by N2 or N2 and slow-wave sleep (SWS) >20 min in duration, which can include wake. The cycle ends with the end of the REM period, which can include wake or short segments of non-REM sleep. No minimum REM duration criterion was applied (*Tarokh et al., 2012*). In some cases (described below), the cycle end was defined at a non-REM sleep stage or wake. Two examples of hypnograms with marked classical sleep cycles are shown in *Figure 1A* – B. Four more examples are presented in *Figure 1—figure supplement 1*.

The last incomplete (not terminated by the REM sleep phase) cycle at the end of the night was included in the analysis if its duration was >50 min. The last incomplete cycles <50 min were removed (nevertheless, they are shown in figures when present).

In Supplementary Excel File shared on https://osf.io/gxzyd, we report classical cycle durations for each participant as scored by two human raters and the automatic algorithm (*Blume and Cajochen, 2021*). In *Appendix 1—table 8*, we report the inter-rater agreement in number and durations of classical cycles.

## Skipped cycles

Given the absence of strict and broadly accepted rules for cycles with skipped REM sleep definition in literature, here, we tagged a cycle as 'skipped' based on the visual inspection of the hypnogram combined with the criteria proposed by *Jenni and Carskadon, 2004* and *Tarokh et al., 2012*. Specifically, we subdivided a long cycle >110 min into two when: (1) there was a 'lightening of sleep' (i.e. the presence of wake, N1 and N2) in the middle of the long cycle, when a REM sleep episode was anticipated, (2) a continuous episode of N1, N2, wake or movement time lasting at least 12 min was preceded and followed by slow-wave sleep *Jenni and Carskadon, 2004*; (3) two clear episodes of slow-wave sleep were separated by lighter non-REM stages (which might include wake; *Campbell et al., 2011*; *Tarokh et al., 2012*). Long cycles containing skipped cycles were divided into cycles at time of sleep lightening. Examples of hypnograms with skipped sleep are shown in *Figure 1—figure supplement 3*. For each dataset, we checked whether the classical cycles with skipped REM sleep had been detected by the fractal cycle algorithm.

In Supplementary Excel File shared on https://osf.io/gxzyd, we report which classical cycles were tagged as 'skipped' by two human raters. In Appendix, we report the inter-rater agreement in number of cycles with skipped REM sleep (*Appendix 1—table 9*). In Supplementary PowerPoint File shared on https://osf.io/gxzyd, hypnograms of all healthy adult participants are presented next to fractal cycles with skipped cycles marked individually as assessed by rater 1.

## Statistical analysis

The assumption that durations of the fractal and classical cycles come from a standard normal distribution was tested using the one-sample Kolmogorov-Smirnov test. The result suggested that this

assumption should be rejected (p<0.05); therefore, non-parametric tests were used for all further analyses.

We correlated fractal and classical cycle durations using Spearman's correlations in each dataset separately as well as in all datasets pooled. Given that in some participants (from 34 to 55% in different datasets), the number of the fractal cycles (mean 4.6±1.0 cycles per participant) was not equal to the number of the classical cycles (mean 4.7±0.9 cycles per participant), prior to the correlation analysis, we averaged the duration of the fractal and classical cycles over each participant. For a subset of the participants (45–66% of the participants in different datasets) with a one-to-one match between the fractal and classical cycles, we performed an additional correlation without averaging, that is, we correlated the durations of individual fractal and classical cycles.

To identify sources of fractal and classical cycle mismatch, we further correlated between the difference in classical vs fractal sleep cycle durations on the one side and either the amplitude of fractal descend/ascend (to reflect fractal cycle depth), duration of cycles with skipped REM sleep, duration of wake after sleep onset or the REM episode length of a given cycle (to reflect peak flatness) on the other side (*Table 2*).

Likewise, we computed person-centered effect sizes, the approach that answers the question, 'How many participants in the study showed the consistent with theoretical expectation effect?'. This approach helps to reveal data patterns that are missed by traditional statistical analyses (*Grice et al., 2020*). We calculated the sample prevalence by counting the number of significant correlations between fractal and classical cycle duration divided by the total number of cases (both significant and non-significant).

To assess the population prevalence of the findings with associated uncertainty, we used the Bayesian prevalence, accounting for the false positive rate of the statistical test (*Ince et al., 2022*). This method helps to estimate the proportion of the population that would show the effect if they were tested in this experiment or, in other words, the population within-participant replication probability (*Ince et al., 2022*). As an output, this method provides the maximum a posterior estimate – the most likely value of the population parameter. To quantify the uncertainty of this estimate, Bayesian prevalence also provides the highest posterior density intervals for various levels (we used the 96% probability level) – the range within which the true population value lies with the specified probability. To perform this analysis, we used an online web application available at https://estimate.prevalence. online.

To compare pediatric and young adult groups (*Appendix 1—table 3*), MDD patients and controls (*Table 3*), MDD patients treated with REM-suppressive antidepressants and patients treated with REM-non-suppressive antidepressants (*Appendix 1—table 5*), we used the non-parametric Mann-Whitney U test. We performed the analyses both at the cycle level (while pooling the cycles of all participants together) as well as at the subject level (while averaging the cycles of a given participant). Given that the results of both analyses were similar, we report only the cycle level analysis for simplicity. To compare medicated and unmedicated states of the MDD patients (*Table 3*), we used the paired samples Wilcoxon test. Effect sizes were calculated with Cohen's d.

In Appendix, we report autocorrelations and partial autocorrelations of fractal slope time series (*Figure 4—figure supplement 7*) as well as cross-correlations (*Figure 4—figure supplement 8*, *Appendix 1—table 4*) between time series of fractal slopes vs. time series of non-REM or REM sleep proportion to further model their temporal relationships.

## Acknowledgements

We acknowledge that the Child Development Center, University Children's Hospital Zürich, University of Zürich is the source of the pediatric data (here, referred to as "Dataset 6"). Namely, we would like to thank Carina Volk, Valeria Jaramillo, Renato Merki and Mirjam Studler for the collection of the pediatric data.

# Additional information

## Funding

| Funder | Grant reference number | Author |
|---|---|---|
| Dutch Research Council | Veni | Martin Dresler |
| Development and Innovation Fund of the Ministry of Innovation and Technology of Hungary | TKP2021- EGA-25 | Róbert Bódizs |
| Development and Innovation Fund of the Ministry of Innovation and Technology of Hungary | ÚNKP-22-3-II | Róbert Bódizs |
| Swiss National Science Foundation | 320030_153387 | Melanie Furrer Reto Huber |
| Swiss National Science Foundation | 320030_179443 | Melanie Furrer Reto Huber |
| HMZ Flagship Grant of the University Medicine Zurich | SleepLoop | Melanie Furrer Reto Huber |

The funders had no role in study design, data collection and interpretation, or the decision to submit the work for publication.

## Author contributions

Yevgenia Rosenblum, Conceptualization, Data curation, Formal analysis, Supervision, Validation, Investigation, Visualization, Methodology, Writing - original draft, Project administration, Writing – review and editing; Mahdad Jafarzadeh Esfahani, Data curation, Writing – review and editing, Acquisition and curation of the data of Dataset 4; Nico Adelhöfer, Paul Zerr, Csenge G Horváth, Bence Schneider, Róbert Bódizs, Writing – review and editing; Melanie Furrer, Data curation, Writing – review and editing, Acquisition and curation of the data of Dataset 6; Reto Huber, Data curation, Writing – review and editing, Acquisition and curation of the data of Dataset 6; Famke F Roest, Investigation, Visual scoring of sleep cycle timing and duration (referred to as "Scorer 2" in Manuscript); Axel Steiger, Data curation, Writing – review and editing, Curation of the data of Datasets 1 - 3 and A - C; Marcel Zeising, Data curation, Writing – review and editing, Acquisition and curation of the data of Datasets 1 - 3 and A - C; Martin Dresler, Resources, Supervision, Funding acquisition, Project administration, Writing – review and editing

## Author ORCIDs

Yevgenia Rosenblum ⬦ https://orcid.org/0000-0001-6792-787X
Martin Dresler ⬦ https://orcid.org/0000-0001-7441-3818

## Ethics

The studies were approved by the Ethics committee of the University of Munich (Datasets 1 - 3, 5), Radboud University (Dataset 4) and Canton of Zürich (Dataset 6). Protocol or reference numbers associated with the ethical approval of each study retrospectively analyzed in the current paper are reported in the original papers that are in turn listed in Table 6. All participants (or participants' parents for Dataset 6) gave written informed consent.

Reviewer #1 (Public review): https://doi.org/10.7554/eLife.96784.4.sa1
Reviewer #2 (Public review): https://doi.org/10.7554/eLife.96784.4.sa2
Author response https://doi.org/10.7554/eLife.96784.4.sa3

# Additional files

## Supplementary files
MDAR checklist

## Data availability

The fractal slopes and sleep stages for each 30-second epoch of sleep for healthy adult participants, Matlab scripts calculating fractal slopes and fractal cycles, Excel file used to perform all the statistical analyses with the fractal and sleep characteristics for each participant and PowerPoint file depicting fractal and classical cycles for all participants can be accessed at https://osf.io/gxzyd.

The following dataset was generated:

| Author(s) | Year | Dataset title | Dataset URL | Database and Identifier |
|---|---|---|---|---|
| Rosenblum Y, Esfahani MJ, Dresler M | 2024 | Fractal cycles of sleep | https://osf.io/gxzyd | Open Science Framework, gxzyd |

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

# Appendix 1

## Participants

### Healthy adults
We retrospectively analyzed polysomnographic recordings from the following studies *Table 1*:

### Datasets 1–3
40, 40, and 33 healthy controls from three independent sleep studies in MDD conducted at the Max Planck Institute of Psychiatry, Germany. These datasets are described in *Rosenblum et al., 2023a* and *Bovy et al., 2022*. In addition, these participants are used as controls in MDD datasets A–C described below.

### Dataset 4
36 healthy participants from a home-based sleep study exploring simultaneous polysomnographic and EEG wearables conducted at the Donders Institute for Brain, Cognition and Behavior, the Netherlands (Described as Dataset 2 in *Jafarzadeh Esfahani et al., 2023*). The signal was recorded at participants' homes over three nights with a gap of a week between each recording. For consistency with other datasets (i.e. to end up with a comparable number of cycles provided by each participant), we used polysomnography (and not EEG recorded by wearables) from the first night only since it had the largest sample size (i.e. 5 subjects dropped out from the study after the first polysomnographic recording).

### Dataset 5
68 healthy controls from previous endocrinological studies conducted at the Max Planck Institute of Psychiatry, Germany, using only nights with no pharmacological or endocrine intervention. 60/68 participants are described in *Rosenblum et al., 2023b*.

### Dataset 6
21 healthy children and adolescents from previous studies (*Furrer et al., 2019*; *Volk et al., 2019*; *Jaramillo et al., 2020*) conducted at the University Children's Hospital Zürich, Switzerland. For the control group to this dataset, we selected all healthy adults from Datasets 1–3, 5, 6 (n=205) whose ages lay in the range of 23–25 years the age when the brain maturation process is supposed to be finished *Giedd and Rapoport, 2010* and no age-related processes are expected to start. This resulted in 24 subjects with a mean age of 24.8±0.9 years (*Appendix 1—table 3* here).

In *Rosenblum et al., 2023a*, Datasets A, B, and C are referred to as the Replication Dataset 2, Main Dataset and Replication Dataset 1, respectively; in *Bovy et al., 2022*, the naming is the same as here.

### Patients with MDD
We retrospectively analyzed polysomnographic recordings from our previous studies (*Bovy et al., 2022*; *Rosenblum et al., 2023a*, *Table 1*; *Table 2*):

### Dataset A
40 long-term medicated MDD patients *vs.* 40 age- and gender-matched healthy controls (Dataset 1 here).

### Dataset B
38 MDD patients in unmedicated and 7-day medicated states *vs.* 40 healthy age and gender-matched controls (Dataset 2 here).

### Dataset C
33 MDD patients at 7 day and 28 day of medication treatment *vs.* 33 healthy age and gender-matched controls (Dataset 3 here).

Demographic and sleep characteristics of the patients, medication treatment and polysomnographic devices are described in our previous works (*Bovy et al., 2022*; *Rosenblum et al., 2023a*). Here, *Appendix 1—table 5* presents medication treatment. All studies were approved by the Ethics committee of the University of Munich. All patients gave written informed consent.

## Excluded participants

Dataset 1: in one participant, the recording was available for the first 90 minonly; in another participant, 25% of epochs were defined as 'wake'; these two participants were excluded from the analysis. Dataset 2: in one participant, the recording was available for the first 133 minutes only; this subject was not included in the analysis. Dataset 3: in one participant 35% of all epochs were defined as 'wake' and their data was excluded. Dataset 4: in one participant 50% of the epochs were tagged as 'wake', another participant had no REM epochs, therefore, his classical sleep cycles could not be defined. These two participants were excluded. Dataset 5: in 5/68 participants, more than 30% of all epochs were tagged as 'wake'. In addition, one participant had no REM epochs. These six participants were excluded from further analyses (*Table 6*). No pediatric and MDD participants were excluded.

Notably, in many sleep studies > 25% of wake is not an exclusion criterion. However, here, we focus on sleep cycles specifically and can not assume that such a prolonged period of wakefulness during a night can be considered a part of a sleep cycle. An example of the data of one excluded participant is given in *Figure 1—figure supplement 3C* (S37).

### Fractal cycles in patients with insomnia: a pilot

We compared fractal cycle duration in 11 patients with insomnia (18.18% male; age: 44±13.2 years, n=11, No. cycles = 51) and 11 healthy controls (54.5% male; age: 42.4±15.4 years, n=11, No. cycles = 46), using open access dataset from a cross-sectional study on psychophysiological insomnia (*Rezaei et al., 2017*). The analysis were performed as described in Methods. However, due to low-pass filtered EEG data, the fitting of spectral slopes was performed in the 1–18 Hz range.

An individual example of smoothed fractal slope time series and hypnograms is shown in in *Figure 3—figure supplement 2A*. We found that patients with insomnia showed a shorter duration of fractal cycles compared to controls with a medium effect size (83±45 *vs* 101±43 min, p=0.04, Cohen's d=–0.4, 4.6 cycles/participant *vs.* 4.2 cycles/participant, *Figure 3—figure supplement 2B, C*).

These findings are in line with the existing literature on flatter slopes in insomnia patients compared to controls (*Andrillon et al., 2020* and *Figure 3—figure supplement 2D*), strengthening the hyperarousal model of insomnia.

This analysis is an outlook only and future studies using a higher sample size should confirm this finding.

**Appendix 1—table 1.** Fractal cycle topography in healthy adults.

| Area | Dataset 1 | Dataset 2 | Dataset 3 | Dataset 4 | Dataset 5 |
|---|---|---|---|---|---|
| Classical sleep cycle duration, min | | | | | |
| --- | 86.2±23.3 | 90.0±21.3 | 89.0±22.7 | 92.2±23.7 | 91.9±29.0 |
| Fractal sleep cycle duration, min | | | | | |
| F | NA | 90.0±25.5 | 86.4±31.2 | 94.7±37.1 | 89.9±37.1 |
| C | 86.4±35.2 | 91.1±29.4 | 85.2±34.2 | 95.4±37.3 | 90.8±39.9 |
| P | NA | 95.0±32.3 | 89.8±37.2 | NA | 90.7±41.9 |
| O | NA | 89.7±28.6 | 85.0±31.3 | 100.0±47.0 | 92.2±42.5 |
| Classical-fractal cycles correlation, r coefficient | | | | | |
| F | NA | 0.508 | 0.565 | 0.474 | 0.513 |
| C | 0.331 | 0.364 | 0.213 | 0.478 | 0.277 |
| P | NA | 0.273 | 0.120 | NA | 0.411 |
| O | NA | 0.306 | 0.239 | 0.516 | 0.279 |
| Classical-fractal cycles correlation, p-value | | | | | |
| F | NA | 0.001 | 0.001 | 0.005 | <0.001 |
| C | 0.042 | 0.023 | 0.242 | 0.004 | 0.029 |

*Appendix 1—table 1 Continued on next page*

*Appendix 1—table 1 Continued*

| Area | Dataset 1 | Dataset 2 | Dataset 3 | Dataset 4 | Dataset 5 |
|------|-----------|-----------|-----------|-----------|-----------|
| P | NA | 0.093 | 0.512 | NA | <0.001 |
| O | NA | 0.058 | 0.189 | 0.002 | 0.028 |

F – frontal (averaged over F3 and F4), C – central (averaged over C3 and C4), P – parietal (averaged over P3 and P4), O – occipital (averaged over O1 and O2) electrodes.

**Appendix 1—table 2.** Fractal cycle characteristics: frequency bands comparison.

| Parameter | 0.3–30 Hz | 1–30 Hz |
|-----------|-----------|---------|
| Fractal cycles, No./night | 4.6±1.0 | 4.6±1.1 |
| Fractal sleep cycle duration, min | 89.1±34.0 | 90.7±36.9 |
| Descent amplitude, z | –2.2±0.8 | –2.1±0.8 |
| Ascent amplitude, z | 2.2±0.6 | 2.1±0.6 |
| Classical – fractal cycles duration correlation, r | 0.488 | 0.397 |
| Classical – fractal cycles duration correlation, p | 10e-13 | 10e-9 |

Mean values ± SD (min) are presented for the pooled dataset (n=205). The 1–30 Hz band is added to control for a possible distortion (the so called 'knees'' of the spectrum) of the linear fit by excluding low frequencies with strong oscillatory activity. Both bands, however, show similar results probably because of the smoothening procedure used in this study.

**Appendix 1—table 3.** Pediatric demographic and sleep characteristics.

| Characteristic | Children and adolescents (Dataset 6) | Controls: Young adults (from Datasets 2, 4, 5) |
|----------------|--------------------------------------|------------------------------------------------|
| Sample size | 21 | 24 |
| Age, years | 12.4+3.1 | 24.8+0.9 |
| Age range, years | 8–17 | 23–25 |
| Wake, % | 4.94 | 6.09 |
| Non-REM stage 1, % | 3.29* | 7.27 |
| Non-REM stage 2, % | 41.89 | 42.23 |
| Slow-wave sleep, % | 31.06 | 24.92 |
| REM sleep, % | 18.82 | 18.58 |
| Total sleep time, min | 444+37 | 441+44 |
| Classical sleep cycle duration, min | 80.4+23.0* | 89.8+22.2 |
| Fractal sleep cycle duration, min | 75.5+33.7* | 94.1+32.1 |
| Ascent amplitude, z | 2.2+0.7 | 2.1+0.6 |
| No. fractal cycles | 112 | 121 |
| No. classical cycles | 112 | 114 |

## Autocorrelation and partial autocorrelation

### Method

To further explore the fluctuating nature of the fractal slope time series, we assessed the autocorrelation and partial autocorrelation patterns of this data using Matlab's *autocorr* and *parcorr* functions, respectively. In autocorrelation, a given value from a time series is regressed on previous values from that same time series. Partial autocorrelation is similar to autocorrelation except that it displays only the correlation between two observations that the shorter lags between those observations do not explain. In other words, the partial correlation for each lag is the unique correlation between those two observations after partialling out the intervening correlations, that is it controls for other lags.

For both autocorrelation and partial autocorrelations, we defined the number of lags as 180, which corresponds to 90 min, an average duration of a fractal cycle, with a 30 s step. We assessed these correlations in each participant separately and then averaged the correlation coefficients for each time lag over all participants of a given dataset and in a pooled dataset.

## Results

Autocorrelation strength decayed throughout the lags showing a somewhat sinusoid shape. Specifically, positive correlations of moderate strength were observed for the 0.5–14 min lags; for the 14–25 min lags, the correlations were weak. In addition, weak positive autocorrelations were observed around the 90th minute while weak negative autocorrelations were observed around the 45th minute (*Figure 4—figure supplement 7*, left), corroborating 90 min periodicity of fractal cycles.

The partial autocorrelation that controls for other lags further revealed that only 0–5 min lag coefficients were statistically significant, that is autocorrelation equals zero at lags greater than 5 min (*Figure 4—figure supplement 7*, right). This finding indicates that the fractal slopes of the consecutive sleep epochs within a given 5 min are not independent, they autocorrelate so that they are much more likely to appear in an observed pattern (e.g. as in *Figure 1A*) than expected by chance.

## Cross-correlations

### Method

To further model the temporal relationship between fractal and classical cycles, we explored them on a finer grained level using cross-correlations between time series of fractal slopes and time series of non-REM or REM sleep proportion per each 5 min of sleep. Cross-correlation allows one to evaluate how two time series might concomitantly covary in the same or opposite directions at given temporal intervals (i.e. lags). Thus, it enables the identification of temporally coordinated fluctuations between two variables. From the shape of the cross-correlation function, information concerning the possible direction of the influence between the two processes can be obtained. Cross-correlation analyses were performed between the time series of the fractal slopes averaged over each 5 min (based on the results of autocorrelation analysis reported above) of sleep on one side and the time series of the proportion of either REM or non-REM stages 2 and 3 (together) on the other side. We averaged the values of all the time series (originally calculated for each 30 s of sleep) over each 5 min of sleep. The cross-correlation coefficients were computed for lags ranging from –40 to 40 min, a period approximately corresponding to an average duration of a fractal cycle, with a 5 min step. Confidence intervals of 95% were calculated to infer statistical significance. We calculated cross-correlations in each participant separately and then averaged correlation coefficients for each time lag over all healthy adult participants in a pooled dataset (Datasets 1–5). Likewise, we calculated the proportion of the participants who showed statistically significant correlation coefficients for each lag.

### Results

We found that the time series of fractal slopes positively cross-correlated with the time series of REM sleep proportion and negatively correlated with the time series of non-REM sleep proportion for lags lying between –15 and +5 min (*Figure 4—figure supplement 8*, *Appendix 1—table 4*). At the individual level, significant correlations were observed in more than 80% of the participants for the –10–0 minute lags. Bayesian prevalence analysis revealed that the maximum a posterior prevalence estimate is equal to 0.79 while the Bayesian highest posterior density interval (true population level) with 96% probability level lies within the 0.73–0.85 range.

**Appendix 1—table 4.** Cross-correlations.

| Time lag | non-REM<-Slopes | Slopes<-non-REM | REM<-Slopes | Slopes<-REM |
|---|---|---|---|---|
| Correlation coefficients, r | | | | |
| 0 min | –0.50 | | 0.40 | |
| 5 min | –0.50 | –0.36 | 0.44 | 0.32 |
| 10 min | –0.44 | n.s. | 0.40 | n.s. |
| 15 min | –0.33 | n.s. | 0.33 | n.s. |

*Appendix 1—table 4 Continued on next page*

*Appendix 1—table 4 Continued*

| Time lag | non-REM<-Slopes | Slopes<-non-REM | REM<-Slopes | Slopes<-REM |
|---|---|---|---|---|
| 20 min | n.s. | n.s. | n.s. | n.s. |
| Participants showing significant effect, % | | | | |
| 0 min | 85 | | 80 | |
| 5 min | 85 | 70 | 87 | 72 |
| 10 min | 84 | 54 | 86 | 53 |
| 15 min | 76 | 21 | 79 | 18 |
| 20 min | 52 | 9 | 5 | 61 |

REM – rapid eye movement sleep, n.s. – non-significant. Only lags associated with statistically significant correlation coefficients are reported, r's higher than 0.7 are considered as strong correlation scores, values lower than 0.3 are considered as weak, r's values in the range of 0.3–0.7 are considered as moderate scores. Correlation coefficients were calculated for each healthy adult individually and then averaged over all participants.

**Appendix 1—table 5.** Demographic and clinical characteristics of the subgroups of patients by medication class (mean ± SD).

| Medication class | n | Age | No. females | No. of previous depressive episodes | HAM-D baseline | HAM-D 7 day |
|---|---|---|---|---|---|---|
| SSRI (citalopram, escitalopram, paroxetine, sertraline) | 13 | 29.9±10.0 | 8 | 0.6±0.8 | 19.7±4.2 | 13.9±4.6 |
| TCA (trimipramine, amitriptyline, amitriptylinoxide) | 8 | 36.6±11.9 | 4 | 2.1±1.1 | 22.1±3.4 | 16.6±5.5 |
| NDRI (bupropion) | 6 | 30.7±10.5 | 3 | 0.7±0.5 | 18.5±3.5 | 17.8±3.2 |
| SNRI (venlafaxine, duloxetine) | 6 | 31.7±10.9 | 2 | 1.7±0.83 | 18.3±2.5 | 14.7±5.5 |
| NaSSA (mirtazapine) | 5 | 26.8±6.1 | 3 | 2.6±3.2 | 20.2±4.8 | 13.8±4.7 |
| REM suppressive (SSRI, SNRI, amitriptyline, amitriptylinoxide) | 21 | 31.1±10.3 | 11 | 1.1±1.0 | 19.2±3.6 | 14.1±4.5 |
| REM non-suppressive (trimipramine, bupropion, mirtazapine) | 17 | 31.6±10.4 | 7 | 1.7±2.0 | 20.7±4.1 | 16.6±4.9 |

HAM-D – Hamilton Depression Rating Scale, REM – rapid eye movement sleep, SD – standard deviation, NaSSA – noradrenergic and specific serotonergic antidepressants, NDRI – norepinephrine-dopamine reuptake inhibitor, SNRI – serotonin-norepinephrine reuptake inhibitors, SSRI – selective serotonin reuptake inhibitors, TCA – tricyclic antidepressants.

**Appendix 1—table 6.** Fractal cycle topography in MDD.

| Dataset | Group | F | C | P | O |
|---|---|---|---|---|---|
| A | Healthy controls | NA | 84±35 | NA | NA |
| | long-termed med. MDD | NA | 97±43* | NA | NA |
| B | Healthy controls | 90±26 | 91±29 | 95±32 | 90±29 |
| | unmed. MDD | 92±38 | 92±39 | 95±44 | 92±35 |
| | 7-d med. MDD | 105±45*,† | 100±49* | 107±49*,† | 105±51*,† |

*Appendix 1—table 6 Continued on next page*

*Appendix 1—table 6 Continued*

| Dataset | Group | F | C | P | O |
|---------|-------|---|---|---|---|
| C | Healthy controls | 88±32 | 86±34 | 90±37 | 85±31 |
| | 7-d med. MDD | 107±48* | 108±49* | 105±46* | 107±44* |
| | 28-d med. MDD | 106±51* | 95±41* | 104±51* | 105±51* |

Mean durations ± SD (min) are presented, MDD – major depressive disorder, unmed. – unmedicated, med. – medicated. F – frontal (averaged over F3 and F4), C – central (averaged over C3 and C4), P – parietal (averaged over P3 and P4), O – occipital (averaged over O1 and O2) electrodes.
*statistically significant p-values of the t-test that compares a given group to age-matched controls.
†statistically significant p-values of the t-test comparing medicated and unmedicated states of MDD patients (dataset B)

## Intra-fractal method reliability

To assess the intra-fractal method reliability, we correlated between the durations of fractal cycles (i.e., the time interval between two adjacent local peaks) calculated as defined in the main text, that is using a minimum peak prominence of 0.94 z and smoothing window of 101 thirty-second epochs, with those calculated using a minimum peak prominence ranging from 0.86 to 1.20 z with a step size of 0.04 z and smoothing windows ranging from 81 to 121 thirty-second epochs with a step size of 10 epochs (*Appendix 1—table 7*). We found that fractal cycle durations calculated using adjacent minimum peak prominence (i.e. those that differed by 0.04 z) showed r's>0.92, while those calculated using adjacent smoothing windows (i.e. those that differed by 10 epochs) showed r's>0.84.

In addition, we correlated fractal cycle durations defined using different channels and found that the correlation coefficients ranged between 0.66–0.67 for all datasets except Dataset 2, which showed lower than expected (while still significant) correlations coefficients in the range of 0.42–0.45 (*Appendix 1—table 1*).

Thus, most of the correlations performed to assess intra-fractal method reliability showed correlation coefficients (*r*>0.6) higher than those obtained to assess inter-method reliability (*r*=0.41–0.55), i.e., correlations between fractal and classical cycle durations (*Table 1* and *Figure 1C* of the main text and *Appendix 1—table 7* here). The strongest correlation between the durations of fractal vs classical cycles (r's>0.45) were obtained while using the minimum peak prominence of 0.94 and 0.98 z-values, smoothing windows of 101 and 111 (i.e. 50.5 and 55.5 min, *Appendix 1—table 7*) and frontal channels (*Appendix 1—table 1*).

**Appendix 1—table 7.** Intra-fractal method reliability.

| Peak prominence variation | | 0.86 z | 0.90 z | 0.94 z | 0.98 z | 1.20 z |
|---|---|---|---|---|---|---|
| | 0.86 z | - | 0.93 | 0.86 | 0.82 | 0.55 |
| | 0.90 z | - | - | 0.92 | 0.88 | 0.60 |
| | 0.94 z | - | - | - | 0.95 | 0.67 |
| | 0.98 z | - | - | - | - | 0.74 |
| | 1.20 z | - | - | - | - | - |
| Correlations, r | Classical cycles | 0.45 | 0.45 | 0.46 | 0.46 | 0.31 |
| | Fractal cycles | 88.6 | 90.5 | 92.7 | 93.8 | 106.1 |
| Duration, min | Classical cycles | 91.0 | | | | |
| **Smoothing window, 30 s epochs** | | 81 | 91 | 101 | 111 | 121 |

*Appendix 1—table 7 Continued on next page*

*Appendix 1—table 7 Continued*

| Peak prominence variation | | 0.86 z | 0.90 z | 0.94 z | 0.98 z | 1.20 z |
|---|---|---|---|---|---|---|
| | 81 | - | 0.85 | 0.69 | 0.59 | 0.49 |
| | 91 | - | - | 0.84 | 0.72 | 0.63 |
| | 101 | - | - | - | 0.87 | 0.78 |
| | 111 | - | - | - | - | 0.85 |
| | 121 | - | - | - | - | - |
| Correlations, r | Classical cycles | 0.32 | 0.37 | 0.45 | 0.45 | 0.39 |
| Duration, min | Fractal cycles | 84.5 | 88.6 | 92.4 | 94.5 | 98.6 |

Fractal cycle durations were calculated using the minimum peak prominence from 0.86 to 1.20 z with the step of 0.04 z and smoothing window from 81 to 121 epochs with the step of 10 thirty-second epochs, r's higher than 0.7 are considered as strong correlation scores, values lower than 0.3 are considered as weak, r's values in the range of 0.3–0.7 are considered as moderate scores, all correlations were statistically significant, therefore p-values are not reported, r – Spearman correlation coefficients.

## Intra-classical method reliability

To assess the intra-classical method reliability, we correlated between the durations or numbers of classical cycles assessed by two independent human scorers and the automatic sleep cycle detection algorithm, namely, the R 'SleepCycles' package. The results are presented in *Appendix 1—table 8*.

In the pooled dataset, the correlation coefficient between classical cycle durations assessed by the two human scorers was 0.8, ranging from 0.7 to 0.9 in different datasets (in literature, r's>0.7 are interpreted as strong correlations). This is consistent with the literature on sleep staging reporting an average inter-rater agreement of ~82.6% (*Rosenberg and Van Hout, 2013*).

In the pooled dataset, correlation coefficients obtained for sleep cycle durations between human raters and the automatic algorithm showed remarkably lower coefficients of around 0.55–0.59, ranging from 0.30 to 0.69 in different datasets ('moderate' correlations). These coefficients were remarkably lower compared to the coefficients obtained between two human scorers ('strong' correlations). The lowest human-automatic inter-rater agreement ($r=0.3$) was observed for Dataset 4, where, notably, the data was collected at participants' homes by participants.

In summary, intra-classical method correlations between classical cycle durations scored by two human raters were stronger than the inter-method correlations between fractal and classical cycle durations, which ranged from 0.41 to 0.55 (r's in the range of 0.3–0.7 are considered moderate correlations). The human-automatic correlation coefficients for classical cycle durations ($r\sim0.57$) lay within the range of the correlation coefficients between fractal and classical cycle durations ($r\sim0.49$). In other words, here, the strength of the intra- and inter-method correlations was comparable.

**Appendix 1—table 8.** Intra-classical method reliability.

| Characteristic | | Scorer | Dataset 1 | Dataset 2 | Dataset 3 | Dataset 4 | Dataset 5 | Pooled dataset |
|---|---|---|---|---|---|---|---|---|
| | | Scorer 1 | 87.8±12.6 | 91.5±11.7 | 90.0±13.3 | 93.3±11.0 | 93.8±15.1 | 91.6±13.2 |
| | | Scorer 2 | 88.2±11.6 | 89.4±9.7 | 91.6±11.9 | 90.0±11.9 | 90.4±14.6 | 89.8±12.3 |
| | Mean ± SD | Automatic | 94.2±17.7 | 93.0±13.0 | 94.7±10.6 | 101.9±17.9 | 101.2±18.7 | 97.4±16.6 |
| | | Scorer 1–2 | 0.777 | 0.913 | 0.715 | 0.687 | 0.859 | 0.810 |
| | | Scorer 1–automatic | 0.669 | 0.621 | 0.530 | 0.447 | 0.584 | 0.593 |
| Classical sleep cycle duration, min | Correlation, r | Scorer 2–automatic | 0.604 | 0.686 | 0.641 | 0.297 | 0.547 | 0.549 |

*Appendix 1—table 8 Continued on next page*

*Appendix 1—table 8 Continued*

| Characteristic | | Scorer | Dataset 1 | Dataset 2 | Dataset 3 | Dataset 4 | Dataset 5 | Pooled dataset |
|---|---|---|---|---|---|---|---|---|
| | | Scorer 1 | 4.4±0.8 | 4.6±0.7 | 4.6±0.6 | 4.7±0.8 | 4.9±0.8 | 4.7±0.8 |
| | | Scorer 2 | 4.2±0.7 | 4.6±0.7 | 4.4±0.6 | 4.7±0.8 | 4.9±0.8 | 4.6±0.8 |
| | Mean ± SD | Automatic | 4.1±0.9 | 4.3±0.9 | 4.4±0.6 | 4.2±1.1 | 4.2±1.0 | 4.2±0.9 |
| | | Scorer 1–2 | 0.828 | 0.928 | 0.755 | 1.000 | 0.986 | 0.922 |
| | | Scorer 1–automatic | 0.816 | 0.793 | 0.755 | 0.781 | 0.629 | 0.713 |
| No. classical cycles | Correlation, r | Scorer 2–automatic | 0.840 | 0.761 | 0.885 | 0.781 | 0.669 | 0.720 |

± shows mean and SD, r – Spearman's correlation coefficient, 'skipped' cycle – a cycle where a rapid eye movement sleep episode is expected to appear except that it does not.

**Appendix 1—table 9.** Skipped cycle human inter-scorer agreement.

| Scorer | Dataset 1 | Dataset 2 | Dataset 3 | Dataset 4 | Dataset 5 | Dataset 6 | Pooled dataset |
|---|---|---|---|---|---|---|---|
| Scorer 1, Number of Skipped Cycles (%) | 5/38 (13%) | 7/39 (18%) | 1/32 (3%) | 19/34 (56%) | 16/62 (26%) | 10/21 (48%) | 58/226 (26%) |
| Scorer 2, Number of Skipped Cycles (%) | 6/38 (15%) | 7/39 (18%) | 3/32 (9%) | 19/34 (59%) | 19/62 (31%) | 10/21 (48%) | 64/226 (28%) |
| Scorer 1 – Scorer 2 Agreement, % | 83 | 100 | 33 | 100 | 84 | 100 | 91 |

