## [Editor Report · eLife Assessment]

This **valuable** study provides a novel method to detect sleep cycles based on variations in the slope of the power spectrum from electroencephalography signals. The method, dispensing with time-consuming and potentially subjective manual identification of sleep cycles, is supported by **solid** evidence and analyses. This study will be of interest to researchers and clinicians working on sleep and brain dynamics.

---

## [Referee Report · Reviewer #1 (Public review)]

In this study, Rosenblum et al introduce a novel and automatic way of calculating sleep cycles from human EEG. Previous results have shown that the slope of the non-oscillatory component of the power spectrum (called the aperiodic or fractal component) changes with sleep stage. Building on this, the authors present an algorithm that extracts the continuous-time fluctuations in the fractal slope and propose that peaks in this variable can be used to identify sleep cycle limits. Cycles defined in this way are termed "fractal cycles". The main focus of the article is a comparison of "fractal" and "classical" (ie defined manually based on the hypnogram) sleep cycles in numerous datasets.

The manuscript amply illustrates through examples the strong overlap between fractal and classical cycle identification. Accordingly, a high percentage (81%) can be matched one-to-one between methods and sleep cycle duration is well correlated (around R = 0.5). Moreover, the methods track certain global changes in sleep structure in different populations: shorter cycles in children and longer cycles in patients medicated with REM-suppressing anti-depressants. Finally, a major strength of the results is that they show similar agreement between fractal and classical sleep cycle length in 5 different data sets, showing that it is robust to changes in recording settings and methods.

The match between fractal and classical cycles is not one-to-one. For example, the fractal method identifies a correlation between age and cycle duration in adults that is not apparent with the classical method.

The difference between the fractal and classical methods appear to be linked to the uncertain definition of sleep cycles since they are tied to when exactly the cycle begins/ends and whether or not to count cycles during fractured sleep architecture at sleep onset. Moreover, the discrepancies between the two are on the order of that found between classical cycles defined manually or via an automatic algorithm.

Overall the fractal cycle is an attractive method to study sleep architecture since it dispenses with time-consuming and potentially subjective manual identification of sleep cycles. However, given its difference with the classical method, it is unlikely that fractal scoring will be able to replace classical scoring directly. By providing a complementary quantification, it will likely contribute to refining the definition of sleep cycles that is currently ambiguous in certain cases. Moreover, it has the potential to be applied on animal studies which rarely deal with sleep cycle structure.

---

## [Referee Report · Reviewer #2 (Public review)]

Summary:

This study focused on using strictly the slope of the power spectral density (PSD) to perform automated sleep scoring and evaluation of the durations of sleep cycles. The method appears to work well because the slope of the PSD is highest during slow-wave sleep, and lowest during waking and REM sleep. Therefore, when smoothed and analyzed across time, there are cyclical variations in the slope of the PSD, fit using an IRASA (Irregularly resampled auto-spectral analysis) algorithm proposed by Wen & Liu (2016).

Strengths:

The main novelty of the study is that the non-fractal (oscillatory) components of the PSD that are more typically used during sleep scoring can be essentially ignored because the key information is already contained within the fractal (slope) component. The authors show that for the most part, results are fairly consistent between this and conventional sleep scoring, but in some cases show disagreements that may be scientifically interesting.

Weaknesses:

The previous weaknesses were well-addressed by the authors in the revised manuscript. I will note that from the fractal cycle perspective, waking and REM sleep are not very dissimilar. Combining these states underlies some of the key results of this study.

---

## [Author Response]

The following is the authors’ response to the previous reviews.

**Reviewer 1:**
Weaknesses:The match between fractal and classical cycles is not one-to-one. For example, the fractal method identifies a correlation between age and cycle duration in adults that is not apparent with the classical method. This raises the question as to whether differences are due to one method being more reliable than another or whether they are also identifying different underlying biological differences. It is not clear for example whether the agreement between the two methods is better or worse than between two human scorers, which generally serve as a gold standard to validate novel methods. The authors provide some insight into differences between the methods that could account for differences in results. However, given that the fractal method is automatic it would be important to clearly identify criteria for recordings in which it will produce similar results to the classical method.

We thank the reviewer for the insightful suggestions. In the revised Manuscript, we have added a number of additional analyses that provide a quantitative comparison between the classical and fractal cycle approaches aiming to identify the source of the discrepancies between classical and fractal cycle durations. Likewise, we assessed the intra-fractal and intra-classical method reliability.

**Reviewer 2:**
One weakness of the study, from my perspective, was that the IRASA fits to the data (e.g. the PSD, such as in Figure 1B), were not illustrated. One cannot get a sense of whether or not the algorithm is based entirely on the fractal component or whether the oscillatory component of the PSD also influences the slope calculations. This should be better illustrated, but I assume the fits are quite good.

Thank you for this suggestion. In the revised Manuscript, we have added a new figure (Fig.S1 E, Supplementary Material 2), illustrating the goodness of fit of the data as assessed by the IRASA method.

The cycles detected using IRASA are called fractal cycles. I appreciate the use of a simple term for this, but I am also concerned whether it could be potentially misleading? The term suggests there is something fractal about the cycle, whereas it's really just that the fractal component of the PSD is used to detect the cycle. A more appropriate term could be "fractal-detected cycles" or "fractal-based cycle" perhaps?

We agree that these cycles are not fractal per se. In the Introduction, when we mention them for the first time, we name them “fractal activity-based cycles of sleep” and immediately after that add “or fractal cycles for short”. In the revised version, we renewed this abbreviation with each new major section and in Abstract. Nevertheless, given that the term “fractal cycles” is used 88 times, after those “reminders”, we used the short name again to facilitate readability. We hope that this will highlight that the cycles are not fractal per se and thus reduce the possible confusion while keeping the manuscript short.

The study performs various comparisons of the durations of sleep cycles evaluated by the IRASA-based algorithm vs. conventional sleep scoring. One concern I had was that it appears cycles were simply identified by their order (first, second, etc.) but were not otherwise matched. This is problematic because, as evident from examples such as Figure 3B, sometimes one cycle conventionally scored is matched onto two fractal-based cycles. In the case of the Figure 3B example, it would be more appropriate to compare the duration of conventional cycle 5 vs. fractal cycle 7, rather than 5 vs. 5, as it appears is currently being performed.

In cases where the number of fractal cycles differed from the number of classical cycles (from 34 to 55% in different datasets as in the case of Fig.3B), we did not perform one-to-one matching of cycles. Instead, we averaged the duration of the fractal and classical cycles over each participant and only then correlated between them (Fig.2C). For a subset of the participants (45 – 66% of the participants in different datasets) with a one-to-one match between the fractal and classical cycles, we performed an additional correlation without averaging, i.e., we correlated the durations of individual fractal and classical cycles (Fig.4S of Supplementary Material 2). This is stated in the Methods, section Statistical analysis, paragraph 2.

There are a few statements in the discussion that I felt were either not well-supported. L629: about the "little biological foundation" of categorical definitions, e.g. for REM sleep or wake? I cannot agree with this statement as written. Also about "the gradual nature of typical biological processes". Surely the action potential is not gradual and there are many other examples of all-or-none biological events.

In the revised Manuscript, we have removed these statements from both Introduction and Discussion.

The authors appear to acknowledge a key point, which is that their methods do not discriminate between awake and REM periods. Thus their algorithm essentially detected cycles of slow-wave sleep alternating with wake/REM. Judging by the examples provided this appears to account for both the correspondence between fractal-based and conventional cycles, as well as their disagreements during the early part of the sleep cycle. While this point is acknowledged in the discussion section around L686. I am surprised that the authors then argue against this correspondence on L695. I did not find the "not-a-number" controls to be convincing. No examples were provided of such cycles, and it's hard to understand how positive z-values of the slopes are possible without the presence of some wake unless N1 stages are sufficient to provide a detected cycle (in which case, then the argument still holds except that its alterations between slow-wave sleep and N1 that could be what drives the detection).

In the revised Manuscript, we have removed the “NaN analysis” from both Results and Discussion. We have replaced it with the correlation between the difference between the durations of the classical and fractal cycles and proportion of wake after sleep onset. The finding is as follows:

“A larger difference between the durations of the classical and fractal cycles was associated with a higher proportion of wake after sleep onset in 3/5 datasets as well as in the merged dataset (Supplementary Material 2, Table S10).” Results, section “Fractal cycles and wake after sleep onset”, last two sentences. This is also discussed in Discussion, section “Fractal cycles and age”, paragraph 1, last sentence.

To me, it seems important to make clear whether the paper is proposing a different definition of cycles that could be easily detected without considering fractals or spectral slopes, but simply adjusting what one calls the onset/offset of a cycle, or whether there is something fundamentally important about measuring the PSD slope. The paper seems to be suggesting the latter but my sense from the results is that it's rather the former.

Thank you for this important comment. Overall, our paper suggests that the fractal approach might reflect the cycling nature of sleep in a more precise and sensitive way than classical hypnograms. Importantly, neither fractal nor classical methods can shed light on the mechanism underlying sleep cycle generation due to their correlational approach. Despite this, the advantages of fractal over classical methods mentioned in our Manuscript are as follows:

(1) Fractal cycles are based on a real-valued metric with known neurophysiological functional significance, which introduces a biological foundation and a more gradual impression of nocturnal changes compared to the abrupt changes that are inherent to hypnograms that use a rather arbitrary assigned categorical value (e.g., wake=0, REM=-1, N1=-2, N2=-3 and SWS=-4, Fig.2 A).

(2) Fractal cycle computation is automatic and thus objective, whereas classical sleep cycle detection is usually based on the visual inspection of hypnograms, which is time-consuming, subjective and error-prone. Few automatic algorithms are available for sleep cycle detection, which only moderately correlated with classical cycles detected by human raters (r’s = 0.3 – 0.7 in different datasets here).

(3) Defining the precise end of a classical sleep cycle with skipped REM sleep that is common in children, adolescents and young adults using a hypnogram is often difficult and arbitrary. The fractal cycle algorithm could detect such cycles in 93% of cases while the hypnogram-based agreement on the presence/absence of skipped cycles between two independent human raters was 61% only; thus, 32% lower.

(4) The fractal analysis showed a stronger effect size, higher F-value and R-squared than the classical analysis for the cycle duration comparison in children and adolescents vs young adults. The first and second fractal cycles were significantly shorter in the pediatric compared to the adult group, whereas the classical approach could not detect this difference.

(5) Fractal – but not classical – cycle durations correlated with the age of adult participants.

These bullets are now summarized in Table 5 that has been added to the Discussion of the revised manuscript.

**Reviewer #1 (Recommendations for the authors):**
The authors have added a lot of quantifications to provide a more complete comparison of classical and fractal cycles that address the points I raised.Regarding, the question of skipped REM cycles: I am not sure the comparison of skipped cycle accuracies between fractal and manual methods makes sense. To make a fair comparison fractal and 2nd scorer classifications should be compared to the same baseline dataset which doesn't seem to be the case since the number of skipped cycles is not the same. Moreover, it's not indicated whether the fractal method identifies any false positive skipped cycles.

Thank you for this comment. In the revised Manuscript, we have reported the number of false positive skipped cycles identified by the fractal algorithm. Likewise, we have added the comparison between the fractal algorithm and the second scorer detection of cycles with skipped REM sleep (Results, the section “Skipped cycles”, last paragraph). The text has been revised as follows:

“Visual inspection of the hypnograms from Datasets 1 – 6 was performed by two independent researchers. Scorer 1 and Scorer 2 detected that out of 226 first sleep cycles 58 (26%) and 64 (28%), respectively, lacked REM episodes. The agreement on the presence of skipped cycles between two human raters equaled 91% (58 cycles detected by both raters out of 64 cycles detected by either one or two scorers). The fractal cycle algorithm detected skipped cycles in 57 out of 58 (98%) cases detected by Scorer 1 with one false positive (which, however, was tagged as a skipped cycle by Scorer2), and in 58 out of 64 (91%) cases detected by Scorer 2 with no false positives.”

Minor pointsI suggest reporting the values of inter-method / inter-scorer correlations with the classical method in the main text since otherwise interpreting the value for fractal vs classical is impossible.

Thank you for this comment. In the revised Manuscript, we have moved this section to the main text (Table 3).

Table 5 + text of discussion: cycle identification based on hypnograms is claimed to be. "based on arbitrary assigned categorical values" the categories are not arbitrary since they correspond to well-validate sleep states, only the number associated it and this does not seem to be very important since it's only for visualization purposes.

Thank you for this comment. In the revised Manuscript, we have removed the phrase “arbitrary assigned“.